# HYSTAR: HYPERNETWORK-DRIVEN STYLE-ADAPTIVE RETRIEVAL VIA DYNAMIC SVD MODULATION

**Yujia Cai**[1]*, **Boxuan Li**[1]*, **Chenghao Xu**[2] , **Jiexi Yan**[1]†
[1]School of Computer Science and Technology, Xidian University, Xi'an, Shaanxi, China
[2]School of Electronic Engineering, Xidian University, Xi'an, Shaanxi, China
{yujiacai,boxuanli,chx}@stu.xidian.edu.cn, jxyan1995@gmail.com

## ABSTRACT

Query-based image retrieval (QBIR) requires retrieving relevant images given diverse and often stylistically heterogeneous queries, such as sketches, artworks, or low-resolution previews. While large-scale vision–language representation models (VLRMs) like CLIP offer strong zero-shot retrieval performance, they struggle with distribution shifts caused by unseen query styles. In this paper, we propose the Hypernetwork-driven Style-adaptive Retrieval (Hystar), a lightweight framework that dynamically adapts model weights to each query's style. Hystar employs a hypernetwork to generate singular-value perturbations ($\Delta S$) for attention layers, enabling flexible per-input adaptation, while static singular-value offsets on MLP layers ensure cross-style stability. To better handle semantic confusions across styles, we design StyleNCE as part of Hystar, an optimal-transport-weighted contrastive loss that emphasises hard cross-style negatives. Extensive experiments on multi-style retrieval and cross-style classification benchmarks demonstrate that Hystar consistently outperforms strong baselines, achieving state-of-the-art performance while being parameter-efficient and stable across styles.

## 1 INTRODUCTION

Query-based image retrieval (QBIR) (Thomee & Lew, 2012) is a core mechanism for accessing visual information at scale. Given a query, the system must rapidly return images aligned with the user's intent from a large database (Li et al., 2022a; 2023), enabling modern image search and many downstream applications (Isinkaye et al., 2015). In practice, user queries are highly diverse and personalized, especially in style (Li et al., 2021b; Johnson et al., 2015). Expressing intent appropriately and making retrieval models adapt flexibly to heterogeneous query styles remain central challenges for QBIR. Prior works have sought to address this through style-invariant representations (e.g., Domain-Aware SE Network; Semi3-Net (Lu et al., 2021; Lei et al., 2020)), cross-modal alignment and domain adaptation (e.g., BDA-SketRet; Adapt and Align; Domain-Smoothing Network (Chaudhuri et al., 2022; Dong et al., 2023; Wang et al., 2021)), and cross-category generalization (e.g., Generalising Fine-Grained SBIR (Pang et al., 2019)). However, these methods are typically designed for style-specific datasets, limiting their scalability.

Furthermore, recent advancements in large-scale vision–language representation models (VLRMs), such as CLIP (Radford et al., 2021; Li et al., 2022b; 2021a), have demonstrated significant capabilities in discrimination and generalization abilities, enabling QBIR through the incorporation of rich semantic priors acquired during pretraining. Nevertheless, performance tends to deteriorate when the query style significantly diverges from those encountered during pretraining (e.g., sketch, cartoon, or artwork), primarily due to the distributional mismatch in the shared embedding space (Sain et al., 2023; Li et al., 2024a). A direct remedy is fine-tuning VLRMs on target retrieval datasets. For large models, full fine-tuning is expensive and prone to catastrophic forgetting (Laurier; Kemker et al., 2018). Parameter-efficient fine-tuning (PEFT) methods, such as LoRA (Hu et al., 2022) and

---

*Equal contribution
†Corresponding author

VPT (Zhou et al., 2022c; Jia et al., 2022; Zhou et al., 2022b), freeze the backbone while introducing a small number of trainable parameters. Despite their efficiency, existing PEFT approaches are static (Chavan et al., 2023), relying on a single parameter set shared by all inputs, limiting adaptation to diverse, unseen styles at inference (Dong et al., 2023; Li et al., 2024a).

To tackle this issue, some methods (Li et al., 2024b; Ge et al., 2025; Yu et al., 2024; Yan et al., 2024) consider pretraining separate style-specific units and selecting or composing them at test time. For instance, VB-LoRA (Li et al., 2024b) utilizes a vectorized LoRA bank to select the top-k modules based on the input. However, such approaches inevitably entail extensive annotation and a large library of style units, struggling with unseen or cross-style scenarios. Recently, a line of work exemplified by FreestyleRet (Li et al., 2024a; Tang et al., 2025) leverages style cluster priors extracted from training data to guide prompt learning(Jia et al., 2022; Zhou et al., 2022c;b), achieving partial style adaptation. Though these methods have shown some effectiveness, they are inherently limited by the styles observed during training and often underperform on out-of-distribution samples. Consequently, developing methods for flexible, data-driven adaptation to unseen styles remains a significant challenge (Wang et al., 2022; Zhou et al., 2022a; Li et al., 2025; Dong et al., 2025).

In this paper, we introduce the Hypernetwork-driven Style-adaptive Retrieval method, Hystar, a framework for flexible and generalized QBIR that dynamically adapts to diverse styles of different queries, including previously unseen ones. Specifically, Hystar extracts query-specific style representations and employs a lightweight modulation mechanism, where a hypernetwork-driven module generates instance-conditioned updates for attention layers. Rather than predicting full low-rank matrices, Hystar predicts only singular-value perturbations ($\Delta S$), which reduces prediction difficulty and improves training stability. In parallel, static learnable singular-value offsets on MLP layers provide stable cross-style calibration. This combination of dynamic adaptation and static robustness enables flexible yet stable style-specific retrieval.

Moreover, we introduce StyleNCE, an optimal-transport-weighted contrastive criterion that performs importance-weighted global matching over positives and negatives, better modeling semantic confusions in cross-style retrieval (Robinson et al., 2020; Jiang et al., 2023). Overall, Hystar couples an adaptable architecture with a difficulty-aware objective, emphasizing cross-style semantic alignment while remaining lightweight and stable.

Our contributions are summarized as follows:

- **Hypernetwork-driven dynamic PEFT.** We introduce a hypernetwork that generates input-conditioned SVD modulations, overcoming the rigidity of static PEFT. It enables fine-grained, per-input adaptation on attention layers while relying on stable, precomputed offsets for MLPs, thereby achieving both flexibility in handling diverse inputs and stability during optimization.

- **Style-adaptive contrastive learning.** We propose **StyleNCE**, an OT-weighted contrastive loss that adaptively emphasizes hard cross-style negatives, leading to more robust retrieval under distribution shift.

- **Strong empirical results.** Extensive experiments on multi-style retrieval and cross-style classification benchmarks demonstrate the effectiveness of Hystar in dynamic PEFT for style adaptation and multimodal generalization, consistently outperforming strong baselines.

## 2 RELATED WORK

**Query-based Image Retrieval.** Query-based image retrieval (QBIR) has long been a central topic in computer vision, with early surveys highlighting its importance for accessing large-scale visual data (Thomee & Lew, 2012). Traditional methods often relied on handcrafted features or shallow models (Li & Li, 2018; Kumar Verma et al., 2019), but recent advances in vision-language representation models (VLRMs), such as CLIP (Radford et al., 2021), have enabled powerful zero-shot retrieval by aligning images and text in a shared semantic space. Nevertheless, these models are sensitive to style variations (e.g., sketches, artworks, or low-resolution previews), leading to degraded performance under distribution shifts (Li et al., 2024a; Qiu et al., 2022).

**Parameter-Efficient Fine-Tuning (PEFT).** To mitigate the cost of adapting large-scale VLRMs, parameter-efficient fine-tuning (PEFT) methods have been proposed. LoRA (Hu et al., 2022) and

VPT (Jia et al., 2022) introduce a small number of trainable parameters while freezing the backbone, achieving efficient adaptation. However, these methods are inherently static, applying the same parameter mapping to all inputs, which limits their ability to generalize to unseen query styles (Ha et al., 2016). More recent approaches, such as FreestyleRet (Li et al., 2024a), leverage style-cluster priors to guide prompt learning and achieve partial style adaptation. Yet, their reliance on observed style clusters hampers performance in truly out-of-distribution scenarios.

**Dynamic Modulation and Hypernetworks.** An alternative line of work explores dynamic adaptation. Some studies pretrain multiple style-specific modules and perform selection at inference (Li et al., 2024b), but this approach requires large annotated style libraries. Others attempt dynamic module composition (Gu et al., 2024), though they are constrained by a fixed bank of predefined modules. Hypernetworks (Ha et al., 2016) offer a promising alternative by generating instance-conditioned parameters, reducing the dependence on explicit style clusters. These insights motivate our design of a dynamic-static hybrid framework that leverages hypernetwork-driven modulation for flexible yet robust multi-style retrieval.

**Contrastive Learning under Distribution Shift.** Contrastive learning (Hadsell et al., 2006; Oord et al., 2018; He et al., 2020; Chen et al., 2020) has been a cornerstone of multimodal retrieval. Standard objectives such as InfoNCE (Oord et al., 2018) treat all negatives equally, which is suboptimal in cross-style scenarios where distinguishing between easy and hard negatives is crucial. Our proposed StyleNCE introduces an optimal-transport (Peyré et al., 2019; Ren et al., 2025b; You et al., 2025; Zhang et al., 2025) formulation to explicitly reweight negatives by difficulty (Robinson et al., 2020; Jiang et al., 2023), enhancing robustness against semantic confusion across styles.

## 3 METHOD

In this section, we first introduce the problem setup of QBIR under diverse style variations in Sec. 3.1, and present the overall pipeline (Fig. 1). The core idea is to enhance the adaptability of singular value modulation through hypernetwork-driven dynamic parameterization. We then detail our hybrid dynamic PEFT mechanism with static singular value modulation in Sec. 3.2. Sec. 3.3 further describes our OT-weighted StyleNCE loss, which leverages optimal transport to enable flexible and efficient optimization across heterogeneous input styles. Finally, Sec. 3.4 introduces the training and inference procedure of our framework. Our framework is conceptually related to VLRM-based retrieval (Radford et al., 2021; Li et al., 2022b; 2021a; Sain et al., 2023), but specifically targets cross-style scenarios.

### 3.1 PROBLEM DEFINITION AND PIPELINE

We study QBIR in the presence of significant style variations, including sketches, paintings, low-resolution images, and text queries. In this setting, queries may originate from styles different from the target gallery (e.g., natural photos vs. sketches or artistic renderings). The goal is to retrieve semantically aligned images despite style discrepancies. Formally, given a query $q$ (possibly from a disjoint style) and a candidate set $\{p_i\}$, we aim to learn an embedding function $f_\theta(\cdot)$ such that positive pairs $(q, p^+)$ are closer in the embedding space than negatives $(q, p^-)$.

Multi-style QBIR poses two main challenges:

1. **Cross-style discrepancies.** Distinct styles differ substantially in appearance and statistics, hindering pretrained models (Radford et al., 2021; Sain et al., 2023; Zhou et al., 2022a).
2. **Static PEFT limitations.** Methods such as VPT (Jia et al., 2022; Zhou et al., 2022b;c) and LoRA (Hu et al., 2022) excel on seen styles but fail to generalize to unseen styles.

To overcome these challenges, our dynamic PEFT framework leverages the pretrained $DINOv2$ backbone (Oquab et al., 2023) to extract style-aware features $\mathbf{z}$. A lightweight hypernetwork maps $\mathbf{z}$ to singular value increments $\Delta S$ for attention layers, while static singular value modulation is applied to MLP layers, enabling style-adaptive yet stable transformations. Training is guided by the OT-weighted StyleNCE loss, which emphasizes hard negatives while maintaining efficient optimization. As shown in Fig. 1, the query is initially routed through a style extraction branch, and the

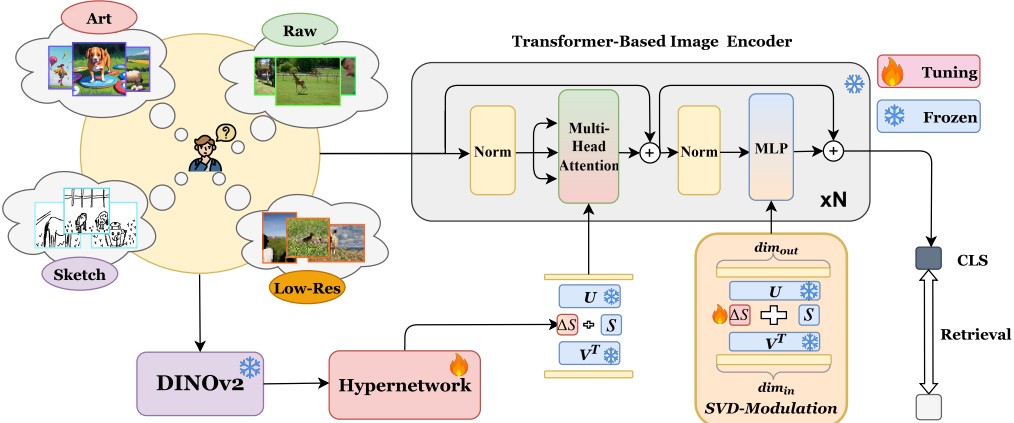

Figure 1: **Overview of the Hystar framework.** For multi-style queries, style features are first extracted using DINOv2. These features are fed into a hypernetwork to produce dynamic singular-value increments for attention layers, enabling style-conditioned modulation of the feature encoder. Additionally, static singular-value increments are applied to the MLP layers, serving as a fixed parameter modulation. Together, these mechanisms guide the encoder to produce style-diverse retrieval predictions.

extracted style is injected into the backbone via Hyper-SVD Modulation. Thereafter, the adapted backbone encodes the query, and the encoded feature is optimized under StyleNCE supervision. This design achieves a balance between computational efficiency and cross-style generalization, significantly boosting retrieval performance under heterogeneous styles.

### 3.2 DYNAMIC–STATIC HYBRID PARAMETER-EFFICIENT FINE-TUNING

We propose a dynamic–static hybrid PEFT method tailored for multi-style adaptation. Let $W_0 \in \mathbb{R}^{d_1 \times d_2}$ denote a pretrained VLRM weight matrix (e.g., attention projections or MLP weights). Our goal is to preserve the spectral structure of $W_0$ while introducing structured, singular value-based updates $\Delta W$ (Lingam et al., 2024; Wang et al., 2025).

To achieve this, we decompose $W_0 \in \mathbb{R}^{d_1 \times d_2}$ as $W_0 = U\Sigma V^\top$, where $U \in \mathbb{R}^{d_1 \times d_1}$ and $V \in \mathbb{R}^{d_2 \times d_2}$ are orthogonal matrices and $\Sigma \in \mathbb{R}^{d_1 \times d_2}$ is diagonal. Rather than learning a full-rank update, we modulate singular values such that $W = W_0 + \Delta W = U(\Sigma + \Delta\Sigma)V^\top$, with $U, V$, and $\Sigma$ frozen and only diagonal increments $\Delta\Sigma$ learned. Using the diagonalization operator $\varphi(\cdot)$, this can be written as $W = U\varphi(s + \Delta s)V^\top$, where $s$ denotes the original singular values and $\Delta s$ their learnable increments.

This preserves pretrained spectra while enabling efficient, stable adaptation. ( A formal justification of the stability of this modulation, based on spectral norm theory, is provided in Appendix A.) Beyond providing stable updates, SVD modulation also offers a geometric advantage for style adaptation. In this formulation, $U$ and $V$ define the semantic subspace of the pretrained model, while $\Sigma$ determines the scaling along these spectral directions. By modulating only the singular values while keeping $U$ and $V$ fixed, the model adapts through smooth, geometry-preserving deformations in the weight manifold. The style-dependent increments $\Delta s$ drive this process, effectively rescaling the intrinsic spectral directions according to the current style. This mechanism aligns the model's representation geometry with style-induced variations, enabling flexible yet stable cross-style adaptation.

Compared with conventional low-rank updates such as LoRA, SVD modulation constrains the update directions within the pretrained spectral subspace. While LoRA introduces new rank components that may interfere with the pretrained semantics, our method only rescales the principal spectral directions, preserving the semantic geometry while adapting their strength to the target style. This spectral alignment enables style-specific flexibility without disrupting the underlying representational structure. Although LoRA introduces no decomposition stage, SVD modulation only

incurs a one-time cost during model initialization for the SVD factorization, after which $U, V$ can be cached and reused. Moreover, the static modulated weights can be merged into the base model for inference, ensuring negligible runtime overhead. In return, the spectral regularity and geometry-preserving adaptation of SVD bring clear advantages in stability and cross-style generalization.

We decompose $\Delta s$ into dynamic and static components:

$$\Delta s = \Delta s_{\text{dyn}} + \Delta s_{\text{stat}}. \tag{1}$$

The static increment $\Delta s_{\text{stat}}$ is globally shared across styles. Initialized at zero to preserve pretrained weights, it evolves during training to capture stable cross-style corrections, ensuring robustness across heterogeneous inputs. A detailed analysis of this dynamic–static decoupling design is provided in Appendix B. The dynamic increment $\Delta s_{\text{dyn}}$ is style-dependent and generated by a hypernetwork conditioned on the style embedding $z \in \mathbb{R}^d$ extracted from the $\texttt{[CLS]}$ token of the $DINOv2$ encoder output, applied to attention layers for cross-style flexibility. Appendix D.3 provides an ablation study on the choice of style extractors. Specifically, the hypernetwork $\mathcal{H}(z; \phi)$ produces

$$\Delta s_{\text{dyn}} = \mathcal{H}(z; \phi) = W_2^{(H)} \sigma\big(W_1^{(H)} z + b_1^{(H)}\big) + b_2^{(H)}, \tag{2}$$

where $\phi$ denotes the learnable parameters $W_1^{(H)}, W_2^{(H)}, b_1^{(H)}, b_2^{(H)}$, and $\sigma(\cdot)$ is a nonlinearity (ReLU/GELU) (Hendrycks & Gimpel, 2023; Nair & Hinton, 2010). The hypernetwork takes the $d$-dimensional style embedding as input and outputs a vector of dimension $r = \min(d_1, d_2)$, corresponding to the number of singular values modulated in each attention projection. Unless otherwise specified, the hidden layer width is set to $2r$. This decomposition allows $\Delta s_{\text{dyn}}$ to adaptively capture cross-style discrepancies, providing *local flexibility* (style adaptation), while $\Delta s_{\text{stat}}$ ensures *global robustness* (task consistency).

By jointly incorporating **style-dependent** and **style-independent** updates in the spectral domain, our approach achieves parameter efficiency while balancing adaptability and robustness, providing a principled new perspective for multi-style image retrieval (Li et al., 2024a; Sain et al., 2023).

### 3.3 STYLeNCE: OT-WEIGHTED CONTRASTIVE LOSS

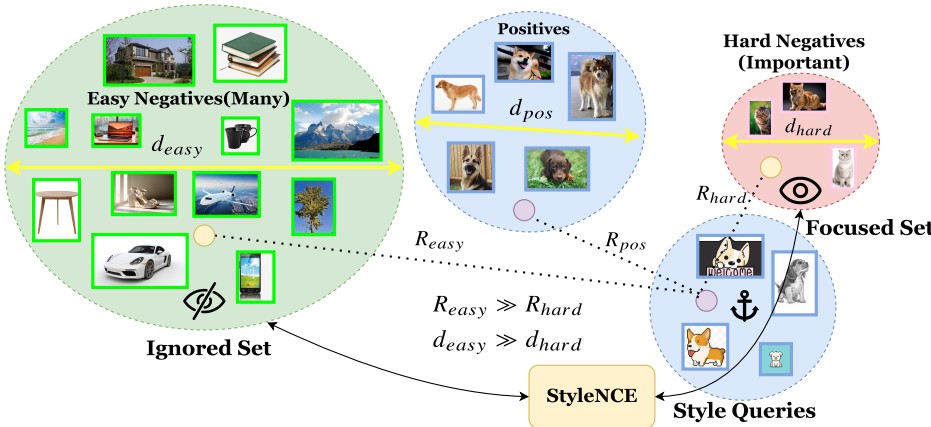

Figure 2: **Motivation behind StyleNCE.** In multi-style queries, most negatives lie far from the query and provide little training signal, whereas hard negatives, though fewer, may be closer to the query than positives due to style-induced abstraction. StyleNCE focuses on these hard negatives, preventing gradients from being dominated by easy samples and ensuring effective optimization for style-diverse retrieval.

Cross-style retrieval suffers from inherent distributional gaps, as sketches emphasize contours, art abstracts geometry, and photos contain complex textures and backgrounds. Standard contrastive losses (e.g., InfoNCE (Oord et al., 2018; He et al., 2020; Chen et al., 2020)) treat all negatives

equally, failing to distinguish between *easy negatives* (semantically distant, offering little supervision) and *hard negatives* (semantically similar yet stylistically different, crucial for generalization). Overemphasis on easy negatives accelerates convergence but weakens out-of-distribution robustness. We propose to reweight negatives by difficulty, amplifying the contribution of hard negatives.

Building on the insight, we design **StyleNCE**, which explicitly reweights negatives according to their difficulty. Given $N$ multi-style queries $Q = [q_1, \ldots, q_N]$ and their ground-truth positives $P = [p_1, \ldots, p_N]$, InfoNCE is:

$$\mathcal{L}_{\text{InfoNCE}} = -\frac{1}{N} \sum_{i=1}^{N} \log \frac{\exp\left(\text{sim}(q_i, p_i)/\tau\right)}{\sum_{j=1}^{N} \exp\left(\text{sim}(q_i, p_j)/\tau\right)}, \tag{3}$$

StyleNCE modifies InfoNCE by introducing difficulty-aware negative weighting (Robinson et al., 2020; Jiang et al., 2023):

$$\mathcal{L}_{\text{StyleNCE}} = -\frac{1}{N} \sum_{i=1}^{N} \log \frac{\exp\left(\text{sim}(q_i, p_i)/\tau\right)}{\exp\left(\text{sim}(q_i, p_i)/\tau\right) + \gamma \sum_{j \neq i} \omega_{ij} \exp\left(\text{sim}(q_i, p_j)/\tau\right)}, \tag{4}$$

where $\gamma$ balances positives vs. negatives, and $\omega_{ij}$ encodes the difficulty of each negative sample.

To compute these difficulty-aware weights, we employ an optimal transport (OT) formulation (Peyré et al., 2019; Ren et al., 2025a; Liu et al., 2025). We define a cost matrix

$$C_{ij} = \begin{cases} \exp((1 - \text{sim}(q_i, p_j))/\lambda), & i \neq j, \\ \infty, & i = j, \end{cases} \tag{5}$$

where $\lambda > 0$ controls emphasis on hard negatives.

We then solve the OT problem

$$\min_{T \in \mathbb{R}^{N \times N}} \langle C, T \rangle, \quad \text{s.t. } T\mathbf{1} = \mathbf{1}, \ T^{\top}\mathbf{1} = \mathbf{1}, \tag{6}$$

where the row- and column-wise normalization constraints explicitly ensure

$$\sum_{i=1}^{N} T_{ij} = 1, \quad \sum_{j=1}^{N} T_{ij} = 1, \quad \forall i, j \in [1, N]. \tag{7}$$

We set the difficulty-aware weights $\omega_{ij} = T_{ij}$ and solve this efficiently using Sinkhorn (Cuturi, 2013) iterations. These OT-based weights systematically amplify hard negatives while maintaining balanced contributions across the batch, allowing StyleNCE to effectively capture cross-style discrepancies.

## 3.4 TRAINING AND INFERENCE

As shown in Figure 1, during training, Hystar first feeds the input image into the style-aware module ($DINOv2$) to obtain a style latent vector $z$. This vector is then passed into the hypernetwork to generate style-specific weight updates for the VLRM attention layers. The updated VLRM encoder processes the input image to produce its embedding. The overall training objective is the proposed StyleNCE loss.

We randomly select a sample from one style in the training style set as the anchor, and use its corresponding ground-truth image as the positive. This encourages the model to learn style-invariant representations while exposing it to diverse style variations.

During inference, embeddings are obtained in the same manner as in training, i.e., by first extracting the style vector $z$, applying the hypernetwork to adjust the VLRM attention weights, and then encoding the input image.

# 4 EXPERIMENT

## 4.1 EXPERIMENTAL SETTINGS

**Implementation Details.** For backbone selection, we primarily use **CLIP (ViT-L/14)** (Radford et al., 2021) and **BLIP (COCO-pretrained)** (Li et al., 2022b), while additional experiments are conducted on **ALBEF (COCO-pretrained)** (Li et al., 2021a), (see Appendix E.2) to demonstrate generality. All trainable models are trained on a single NVIDIA A6000 GPU. For our model, unless otherwise specified, we use a batch size of 48 and train for 35 epochs. We use a learning rate of $1 \times 10^{-3}$ for the static modulation branch and $1 \times 10^{-5}$ for the dynamic hypernetwork to ensure stable training. An ablation on the width and depth of the hypernetwork is provided in Appendix D.2. Dynamic style modulation is injected into the 4th, 7th, 10th, and 13th layers of the CLIP and BLIP ViT backbones. Details on the style injection positions are provided in Appendix D.1. The hyperparameters of the StyleNCE loss are fixed as $\gamma = 80$ and $\lambda = 1$. The Sinkhorn algorithm is run with a sufficiently large number of iterations (50) to ensure convergence. For additional analysis on the effect of varying these parameters, see Appendix G. All inputs follow the default preprocessing pipeline of the selected vision-language representation backbone.

**Datasets.** To validate the effectiveness of our proposed approach, we conduct systematic experiments on the **DSR** (Li et al., 2024a) and **DomainNet** (Peng et al., 2019) datasets. Fine-grained style-level retrieval on DSR serves as our primary benchmark, and we further evaluate on Domain-Net for zero-shot category retrieval and cross-style classification to assess generalization to unseen styles.

**Baseline Methods.** Baselines include CLIP, BLIP, VPT, LoRA, (IA)[3] (Liu et al., 2022), Adapt-Former (Chen et al., 2022), SSF (Lian et al., 2023), FreestyleRet-CLIP, FreestyleRet-BLIP, as well as recent multimodal models ImageBind (Girdhar et al., 2023) and LanguageBind (Zhu et al., 2023b). (* represents the prompt-tuning version of the vanilla models. ) Following the FreestyleRet evaluation protocol, we pretrain VPT, LoRA, (IA)[3], AdaptFormer and SSF on DSR, while FreestyleRet methods are evaluated using their official pretrained weights. The ImageBind and LanguageBind results are adopted from FreestyleRet and are not included in all subsequent experiments.

## 4.2 QUANTITATIVE RESULTS

**Fine-grained Diverse Style Retrieval (DSR).** We first evaluate our approach on the **DSR** dataset, which contains fine-grained categories across diverse query styles. As shown in Table 1, baseline models such as CLIP and BLIP exhibit substantial performance degradation when faced with large style discrepancies. For instance, CLIP only achieves 47.5% Top-1 on Sketch and 45.0% on Low-Resolution queries. Parameter-efficient tuning methods like LoRA, VPT, (IA)[3], AdaptFormer and SSF partially mitigate this issue, improving Top-1 performance to above 70% on Sketch and 80% on Low-Resolution. Retrieval-oriented methods such as FreestyleRet-CLIP and FreestyleRet-BLIP achieve stronger results, surpassing 80% Top-1 on Sketch and 90% on Low-Resolution.

Our proposed Hystar consistently outperforms all baselines across all styles. For instance, Hystar-BLIP achieves Top-1 accuracies of 75.6% on Art, 91.0% on Sketch, and 98.8% on Low-Resolution, yielding absolute improvements of roughly 21–25% over BLIP* in these challenging settings. Notably, it also brings consistent gains on text queries, despite not being explicitly trained for them, demonstrating robustness to both style variation and language-driven retrieval.

Overall, these results highlight that Hystar effectively bridges severe style gaps and sets a new state-of-the-art on fine-grained diverse style retrieval.

**Cross-style Generalization (DomainNet).** We further evaluate our method's generalization to unseen styles via zero-shot retrieval on the DomainNet dataset. As shown in Table 2, Hystar achieves the best zero-shot retrieval performance on the coarse-grained benchmark, surpassing all baselines across both moderate and highly abstract styles, highlighting its strong cross-style generalization.

In contrast, existing approaches exhibit clear limitations in unseen styles. This suggests that the static methods and the style-specific clustering priors leveraged by FreestyleRet may not generalize

| Query Style / Method | Art | | Sketch | | Low-Res | | Text | |
|---|---|---|---|---|---|---|---|---|
| | Top-1 | Top-5 | Top-1 | Top-5 | Top-1 | Top-5 | Top-1 | Top-5 |
| CLIP | 58.5 | 93.7 | 47.5 | 77.3 | 45.0 | 75.7 | 66.1 | 94.7 |
| CLIP* | 58.2 | 90.4 | 63.6 | 93.6 | 78.8 | 97.1 | 72.2 | 96.4 |
| BLIP* | 51.1 | 85.3 | 67.1 | 90.9 | 77.2 | 95.8 | 74.3 | 95.3 |
| LoRA | 63.8 | 96.5 | 72.8 | 96.5 | 79.7 | 95.1 | 70.4 | 97.1 |
| VPT | 66.7 | 96.5 | 73.3 | 97.0 | 81.4 | 96.0 | 69.9 | 96.1 |
| $(IA)^3$ | 64.3 | 96.8 | 71.8 | 95.7 | 80.9 | 96.1 | 70.1 | 96.6 |
| AdaptFormer | 65.1 | 97.0 | 73.5 | 96.4 | 81.1 | 96.3 | 69.7 | 95.8 |
| SSF | 64.7 | 96.4 | 73.0 | 97.0 | 79.9 | 95.8 | 70.1 | 96.3 |
| ImageBind | 58.2 | 86.3 | 50.8 | 79.4 | 79.0 | 96.7 | 71.0 | 95.5 |
| LanguageBind | 67.5 | 92.9 | 63.6 | 89.1 | 78.6 | 94.5 | 79.7 | 98.1 |
| FreestyleRet-CLIP | 71.4 | 97.8 | 80.6 | 97.4 | 86.4 | 97.9 | 69.9 | 97.0 |
| FreestyleRet-BLIP | 74.5 | 97.4 | 81.2 | 97.1 | 90.5 | 98.5 | 81.6 | 99.2 |
| Hystar-CLIP(Ours) | 75.2 | 97.9 | 90.2 | 99.3 | 98.0 | 99.4 | 70.9 | 97.5 |
| Hystar-BLIP(Ours) | **75.6** | **98.1** | **91.0** | **99.8** | **98.8** | **99.9** | **82.0** | **99.6** |

Table 1: **Retrieval performance on the style-diverse QBIR task.** We evaluate Top-1 and Top-5 accuracy(%) on the DSR fine-grained benchmark. The two forms of our Hystar framework, Hystar-CLIP and Hystar-BLIP, outperform in multiple scenarios with different query styles compared with other baselines. Best results are highlighted in **bold**.

| Query Style / Method | Clipart | | Sketch | | Painting | | Quickdraw | | Infograph | |
|---|---|---|---|---|---|---|---|---|---|---|
| | Top-1 | Top-5 | Top-1 | Top-5 | Top-1 | Top-5 | Top-1 | Top-5 | Top-1 | Top-5 |
| CLIP | 60.9 | 77.0 | 49.1 | 67.0 | 59.2 | 75.2 | 9.1 | 15.8 | 41.2 | **60.3** |
| LoRA | 63.0 | 74.8 | 54.6 | 66.8 | 54.8 | 67.2 | 13.1 | 21.9 | 28.3 | 40.4 |
| VPT | 71.7 | 81.6 | 62.5 | 73.8 | 61.7 | 73.3 | 14.5 | 22.6 | 40.6 | 54.9 |
| FreestyleRet | 69.5 | 80.3 | 60.5 | 73.5 | 63.7 | 75.7 | 12.2 | 18.7 | 43.1 | 58.7 |
| Hystar(Ours) | **75.7** | **86.4** | **65.8** | **78.1** | **65.5** | **78.3** | **19.0** | **29.9** | **43.7** | 59.3 |

Table 2: **Zero-shot retrieval performance on unseen styles.** We evaluate Top-1 and Top-5 accuracy(%) on the DomainNet coarse-grained benchmark. Our proposed Hystar framework demonstrates strong performance in zero-shot category-level retrieval under unseen style conditions. Best results are highlighted in **bold**.

consistently to unseen styles, whereas the dynamic multi-style adaptation of our method maintains robust performance.

Overall, these findings demonstrate that while static adaptation methods and style-specific priors struggle to generalize, our methods robustly bridge style gaps and establish SOTA performance in zero-shot category-level retrieval.

**DomainNet Classification Performance Analysis.** To validate the generalizability of our method, we performed zero-shot classification on the DomainNet dataset. Table 3 reports the zero-shot classification performance of various methods on DomainNet across six styles. Although our main focus is multi-style retrieval, this experiment demonstrates the generalization capability of the proposed Hystar model. Hystar consistently achieves the highest accuracy in all styles, outperforming CLIP, LoRA, VPT, and FreestyleRet, particularly in challenging styles such as Sketch (71.2% vs. 64.9%) and Quickdraw (22.9% vs. 14.6%).

| Method | Real | Clipart | Sketch | Painting | Quickdraw | Infograph |
|---|---|---|---|---|---|---|
| CLIP | 82.4 | 72.8 | 64.9 | 68.1 | 14.6 | 53.0 |
| LoRA | 62.4 | 51.7 | 43.1 | 42.8 | 12.8 | 26.1 |
| VPT | 79.9 | 69.8 | 61.9 | 59.2 | 15.7 | 46.0 |
| FreestyleRet | 84.9 | 74.1 | 67.3 | 68.8 | 15.9 | 54.1 |
| Hystar(Ours) | **85.7** | **79.5** | **71.2** | **71.0** | **22.9** | **54.6** |

Table 3: **Zero-shot classification performance of different methods on DomainNet across six styles.** Accuracy (%) for each style is reported, with the best results highlighted in **bold**.

These results indicate that the representations learned by Hystar are not only effective for retrieval cvalso transferable to zero-shot classification, highlighting the model's ability to handle diverse visual styles without additional supervision.

## 4.3 QUALITATIVE RESULTS

(a)CLIP      (b)FreestyleRet      (c)Hystar

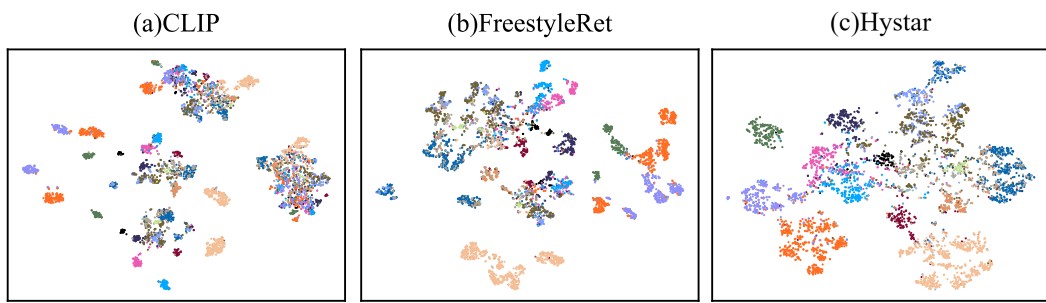

Figure 3: t-SNE visualization of feature embeddings derived by different methods on the DSR dataset. (a)**CLIP**: scattered, overlapping clusters. (b)**FreestyleRet**: more compact but some interclass entanglement. (c)**Hystar**: clearly separable, compact clusters, showing strong cross-style alignment.

To further investigate how different models capture cross-style semantics, we visualize the learned embeddings on the DSR dataset using t-SNE, as shown in Figure 3. Figure 3(a) illustrates that **CLIP** embeddings form scattered and overlapping clusters, reflecting poor discrimination across styles. Figure 3(b) shows that **FreestyleRet** improves cluster compactness, yet several categories remain entangled, suggesting limited separation. In contrast, Figure 3(c) demonstrates that our proposed **Hystar** produces well-structured and clearly separable clusters, with minimal overlap between different styles. These results indicate that Hystar not only enhances retrieval accuracy quantitatively but also achieves qualitatively superior representation learning by aligning cross-style features into more coherent semantic manifolds.

## 4.4 ABLATION STUDY

**Ablation of the Framework Components.** In this section, we perform ablation studies to assess the contributions of the components of Hystar, namely StyleNCE, hypernetwork-driven modulation, and static modulation. As shown in Table 4, the CLIP baseline performs poorly on style-variant queries. Both the Static-only and Hyper-only modulation modules contribute to improved retrieval performance. The Static module provides a globally robust adaptation, effectively aligning representations across different styles in a stable manner; however, it lacks fine-grained flexibility for style-specific adjustments. In contrast, the hypernetwork-based dynamic modulation generates query-specific style adjustments, offering more flexible and personalized adaptation. While this flexibility allows Hyper to capture subtle style variations, it introduces some instability, limiting its standalone improvement.

| Method | Art | Sketch | Low-Res | Text |
|---|---|---|---|---|
| CLIP | 58.5 | 47.5 | 45.0 | 66.1 |
| CLIP + Static | 65.7 | 77.0 | 83.7 | 69.2 |
| CLIP + Hyper | 63.8 | 70.6 | 76.3 | 66.5 |
| CLIP + Hyper + Static | 70.2 | 85.3 | 94.2 | 69.7 |
| CLIP + Hyper + Static + StyleNCE | **75.2** | **90.2** | **98.0** | **70.9** |

Table 4: **Ablation study on the DSR benchmark.** Unless otherwise specified, all variants are trained with standard InfoNCE loss. We report top-1 retrieval accuracy (%) across four query styles: Art, Sketch, Low-Resolution, and Text. Best results are highlighted in **bold**.

| Method | Parameters(M) | Additional Params (%) | Speed(ms) | Inference Time Increase (%) |
|---|---|---|---|---|
| CLIP | 427 | – | 68 | – |
| VPT | 428 | 0.2 | 73 | 7.4 |
| $(IA)^3$ | 427 | 0.1 | 71 | 2.9 |
| AdaptFormer | 429 | 0.5 | 74 | 8.8 |
| FreestyleRet | 476 | 11.5 | 96 | 41.2 |
| Hystar(Ours) | 442 | 3.5 | 108 | 58.8 |

Table 5: **Computation comparison between our Hystar and representative baselines.** For fairness, we only compare CLIP-based models; The percentages of additional parameters and inference-time increase are reported with respect to the CLIP baseline.

These ablations show the two modules are complementary: the Static module provides stable global alignment, while the Hyper module adds query-specific flexibility, and their combination yields clear synergistic gains. Incorporating StyleNCE further improves performance by mining hard style-negatives and boosting cross-style discriminability. Overall, static singular-value modulation secures robust cross-style adaptation, dynamic hypernetwork modulation further improves it, and StyleNCE strengthens handling of style diversity and hard cases—together confirming Hystar's effectiveness for multi-style retrieval under diverse queries.

**Computation Comparison.** We analyze the computational complexity of our framework compared with other baselines. As shown in Table 5, Hystar achieves competitive efficiency among recent retrieval models. Although it introduces a modest increase in inference latency compared with CLIP, VPT, $(IA)^3$, AdaptFormer, and FreestyleRet, the overall parameter scale remains lightweight and only slightly larger than standard vision–language encoders. This marginal computational overhead mainly stems from the controller-guided hypernetwork module, which dynamically modulates representations for style adaptation. Importantly, the trade-off between adaptability and efficiency is well controlled: Hystar substantially improves robustness to visual–textual style variation while preserving a compact and deployable architecture suitable for real-time retrieval scenarios.

## 5 CONCLUSION

In this paper, we present Hystar, a dynamic multi-style retrieval framework that combines hypernetwork-driven dynamic modulation with static singular-value calibration, achieving a balance between adaptability and stability in parameter-efficient fine-tuning. To better handle difficult negatives and style discrepancies, we introduce the OT-weighted StyleNCE loss. Extensive experiments on DSR and DomainNet show that Hystar consistently outperforms strong baselines, while ablations confirm the complementary benefits of dynamic and static modulation. Furthermore, retrieval and classification experiments on unseen styles demonstrate that our method improves the performance of VLRMs under multi-style queries and ensures strong generalization to previously unseen styles. Hystar highlights the effectiveness of dynamic PEFT for style diversity and cross-style generalization, providing a solid foundation for multimodal applications robust to heterogeneous inputs.

## REPRODUCIBILITY STATEMENT

We have taken steps in this work to ensure the reproducibility of our results. All datasets used in our experiments are publicly available. In the main paper and appendices, we provide complete details of all experimental setups, including model architectures, training and evaluation protocols, and hyperparameters. All random seeds are fixed, ensuring that others can replicate our results with the provided code. We believe that the measures we have taken to ensure reproducibility will facilitate straightforward replication and verification of our findings, as well as allow the community to build upon our results in the future.

## ACKNOWLEDGEMENTS

Our work is supported in part by the National Key R&D Program of China (No. 2023YFC3305600), the Joint Fund of Ministry of Education of China (8091B022149, 8091B02072404), National Natural Science Foundation of China (62132016, 62571412, 62302372), and Xidian University Specially Funded Project for Interdisciplinary Exploration (TZJHF202509).

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

APPENDIX

Due to the space limitations of the main text, we provide additional results, ablations, and analysis in this supplementary material. The appendix is organized as follows:

## A  THEORETICAL INTUITION OF SINGULAR-VALUE MODULATION

In this section, we provide a theoretical justification for our design choice of singular-value modulation. In our design, the Hypernetwork predicts low-rank updates by modulating the singular values of LoRA weight matrices, instead of predicting the full weight increment. Let $W_0 \in \mathbb{R}^{d_1 \times d_2}$ denote a pretrained weight matrix, and let its singular value decomposition be

$$W_0 = U\Sigma V^\top, \quad \Sigma = \mathrm{diag}(s_1, \ldots, s_r),$$

where $r = \min(d_1, d_2)$. Our Hypernetwork outputs a singular value increment $\Delta\Sigma$, resulting in the updated weight

$$W = U(\Sigma + \Delta\Sigma)V^\top.$$

The key insight is that the spectral norm (largest singular value) of the update is directly controlled:

$$\|W - W_0\|_2 = \|\Delta\Sigma\|_2 = \max_i |\Delta s_i|.$$

In other words, by modulating only the singular values, we bound the maximum amplification along any input direction, avoiding gradient explosion or collapse. Compared with predicting the full matrix increment $\Delta W$, this approach is both computationally efficient (requiring far fewer parameters) and stable in training.

This theoretical intuition motivates our engineering choice: singular-value modulation ensures that the Hypernetwork can provide style-adaptive updates in a controlled and lightweight manner, which is crucial for stable multi-style retrieval.

| Method | Art | Sketch | Low-Res | Text |
|---|---|---|---|---|
| Dynamic-All Design | 63.7 | 69.9 | 74.8 | 66.3 |
| Reversed Design | 64.1 | 71.2 | 80.2 | 69.1 |
| Original Design | **75.2** | **90.2** | **98.0** | **70.9** |

Table 6: **Comparison of different dynamic modulation strategies in Hystar,** including the **original** design (dynamic-attention and static-MLP), the **reversed** design (dynamic-MLP and static-attention), and the **fully dynamic** design (dynamic-attention and dynamic-MLP). The table reports Top-1 accuracy (%) for joint retrieval with different style queries combined with text. Best results are highlighted in **bold**.

## B    ANALYSIS FOR DYNAMIC ATTENTION AND STATIC MLP DESIGN

To verify the design rationale of employing dynamic modulation in the attention layers while keeping the MLP layers static, we construct a reversed variant of Hystar. In this variant, all attention layers are statically fine-tuned, whereas dynamic modulation is injected into the first linear layer of the MLP blocks at the same layer indices as the original dynamic-attention setup. This design ensures comparable parameter counts while allowing the second linear layer to be indirectly affected. We also include an additional variant where both the attention and MLP layers are dynamically modulated at the same layer indices as in the original design. However, this setting introduces a larger number of trainable parameters, making it incomparable with the other variants under a fixed parameter budget. Moreover, applying static tuning to all layers would contradict the purpose of our style-conditioned modulation and is therefore not considered.

As shown in Table 6, the original design achieves the highest Top-1 accuracy across all style domains, demonstrating that dynamically adapting attention is most effective for cross-style generalization. In contrast, the Dynamic-All configuration performs the worst, indicating that applying dynamic modulation to all layers introduces excessive flexibility and leads to unstable training. The reversed design also underperforms, suggesting that dynamic modulation in MLPs cannot effectively capture style variations. Conceptually, attention layers govern cross-token relationships that are directly influenced by style cues such as texture and composition, making them well-suited for dynamic modulation. In contrast, MLP layers primarily refine semantic features within tokens; keeping them static preserves semantic stability and prevents overfitting to transient style attributes. These results confirm that our hybrid strategy, which employs dynamic attention for style adaptation and static MLPs for semantic stability, achieves the best balance between adaptability and stability.

## C    CROSS-MODAL AND CROSS-STYLE RETRIEVAL RESULTS

| Model | Art+Text | Sketch+Text | Low-Res+Text | Text |
|---|---|---|---|---|
| CLIP* | 57.8(-14.4) | 65.0(-7.2) | 84.7(+12.5) | 72.2 |
| LoRA | 72.2(+1.8) | 79.3(+8.9) | 84.7(+14.3) | 70.4 |
| VPT | 70.6(+0.7) | 79.0(+9.1) | 84.3(+14.4) | 69.9 |
| FreestyleRet | 76.6(+6.7) | 82.5(+12.6) | 86.7(+16.8) | 69.9 |
| Hystar(Ours) | **79.9(+9.0)** | **91.6(+20.7)** | **98.2(+27.3)** | 70.9 |

Table 7: **Top-1 accuracy (%) for joint retrieval with different style queries combined with text.** Values in parentheses indicate the accuracy gain over text-only queries. Best results are highlighted in **bold**.

This section presents the quantitative performance of our model on joint style–text retrieval tasks, followed by a breakdown across different query styles. Following the evaluation protocol established in FreestyleRet, we compute the similarity between each query (style or text) and all gallery images, and adopt the maximum similarity as the final retrieval score.

Table 7 reports top-1 accuracy for three joint query modes (Art+Text, Sketch+Text, Low-Res+Text). On Art+Text, our method achieves 79.9%, surpassing FreestyleRet (76.6%), CLIP (57.8%), as well as parameter-efficient tuning baselines such as LoRA (72.2%) and VPT (70.6%). For Sketch+Text, the advantage becomes more pronounced: our model obtains 91.6%, substantially outperforming FreestyleRet (82.5%) and CLIP (65.0%). Under the Low-Res+Text setting, our method reaches 98.2%, clearly ahead of FreestyleRet (86.7%) and CLIP (84.7%).

Compared to text-only queries, incorporating style queries provides significant performance gains: +9.0% (Art), +20.7% (Sketch), and +27.3% (Low-Res), all markedly higher than those achieved by baseline methods. Moreover, relative to style-only queries (Table 1), our joint approach also yields further improvements (+4.7% on Art, +1.4% on Sketch, and +0.2% on Low-Res). These results suggest that while the marginal benefit of adding text diminishes when style-only queries are already strong, the consistent improvements across all settings highlight the complementary nature of textual and stylistic cues, underscoring the robustness of our multimodal integration strategy.

## D  ADDITIONAL ABLATION OF HYSTAR

In this section, we conduct additional ablation studies to better understand the design choices of Hystar. We systematically analyze how different architectural and functional components contribute to performance and efficiency. Specifically, we examine three key aspects: (1) the selection of injection layers for dynamic modulation, (2) the width and depth configuration of the hypernetwork, and (3) the choice of style feature extractors used for conditioning. Together, these studies provide deeper insights into the trade-offs between adaptability, stability, and computational cost in our framework.

### D.1  ABLATION ON INJECTION LAYERS

Table 8 presents an ablation study on the choice of hypernetwork injection layers across three major query styles (Art, Sketch, and Low-Resolution). The upper block shows results for various middle-layer injection schemes, which constitute our main design. We observe that selectively injecting into middle layers (e.g., $\{4, 6, 8, 10, 12\}$ or $\{4, 7, 10, 13\}$) consistently yields the best trade-off between retrieval accuracy and parameter overhead. Injecting into all intermediate layers ($\{4, 5, 6, \ldots, 12\}$) yields marginal performance improvements but incurs a substantial increase in parameter cost. Moreover, the overly aggressive style-aware adaptation introduces instability, leading to performance degradation compared to configurations with fewer injected layers.

| Injection Layers | Art | Sketch | Low-Res | Top-1 Avg | $\Delta$ Params |
|---|---|---|---|---|---|
| *Middle-layer injection (main study)* | | | | | |
| $\{4, 5, 6, \ldots, 12\}$ | 69.3 | 76.5 | 84.7 | 76.8 | +33.1M |
| $\{4, 6, 8, 10, 12\}$ | **75.4** | 90.1 | **98.6** | **88.0** | +18.4M |
| $\{4, 7, 10, 13\}$ | 75.2 | **90.2** | 98.0 | 87.8 | +14.7M |
| $\{4, 8, 12\}$ | 71.4 | 88.3 | 96.8 | 85.5 | +11.0M |
| *Additional evidence: early / late layers* | | | | | |
| $\{1\}$ | 60.2 | 72.4 | 76.3 | 69.6 | +3.7M |
| $\{16, 19, 22, 24\}$ | 63.1 | 75.4 | 81.7 | 73.4 | +14.7M |

Table 8: **Ablation study on hypernetwork injection layers across three major query styles** (Art, Sketch, and Low-Resolution). We report Top-1 accuracy (%) for each style, averaged performance, and parameter overhead ($\Delta$ Params). Best results are highlighted in **bold**.

The lower block provides additional evidence on early and late layers. Injection into the very early layer ($\{1\}$) results in poor performance, indicating that low-level features captured in early layers are less relevant for style-aware retrieval. Similarly, injecting exclusively into late layers ($\{16, 19, 22, 24\}$) produces only moderate gains, suggesting that later layers are dominated by high-level semantic features, leaving limited room for style modulation. Overall, this ablation confirms

that the middle layers are the most effective region for hypernetwork injection, balancing accuracy improvement and parameter efficiency.

| Network Architecture | Art | Sketch | Low-Res | Top-1 Avg | $\Delta$ Params |
|---|---|---|---|---|---|
| *Width Ablation* | | | | | |
| $\{768 \rightarrow 1024 \rightarrow 1024\}$ | 70.7 | 84.3 | 90.2 | 81.7 | +1.8M |
| $\{768 \rightarrow 2048 \rightarrow 1024\}$ | 75.2 | 90.2 | 98.0 | 87.8 | +3.7M |
| $\{768 \rightarrow 4096 \rightarrow 1024\}$ | **75.4** | **90.8** | **98.6** | **88.3** | +7.3M |
| *Depth Ablation* | | | | | |
| $\{768 \rightarrow 1024\}$ | 68.6 | 80.9 | 88.4 | 79.3 | +0.8M |
| $\{768 \rightarrow 2048 \rightarrow 1024\}$ | **75.2** | 90.2 | **98.0** | **87.8** | +3.7M |
| $\{768 \rightarrow 2048 \rightarrow 2048 \rightarrow 1024\}$ | 74.9 | **90.4** | 97.6 | 87.6 | +7.9M |

Table 9: **Ablation study on hypernetwork layer configurations.** We vary the width and depth of the hypernetwork that predicts dynamic modulation parameters for attention layers. The input dimension (768) corresponds to the feature output of the $DINOv2$ encoder, and the target dimension (1024) corresponds to the singular-value dimension of CLIP's attention weights. All intermediate linear layers are followed by a ReLU activation (omitted in notation for brevity), except for the final projection layer. We evaluate Top-1 accuracy (%) across three major query styles (Art, Sketch, and Low-Resolution), along with the averaged performance and parameter overhead ($\Delta$ Params). Best results are highlighted in **bold**.

## D.2 Ablation on the Width and Depth of the Hypernetwork

The results in Table 9 show clear trends regarding the structural design of the hypernetwork. Increasing the hidden width consistently improves accuracy across all query styles, but the gain saturates beyond a moderate expansion. Specifically, moving from 1024 to 2048 hidden units yields a large performance boost, whereas further enlarging to 4096 offers only marginal improvement at the cost of nearly doubling the parameters. This suggests that excessive width leads to over-parameterization with diminishing returns. Regarding depth, extending the hypernetwork beyond two layers does not improve accuracy and can even cause slight degradation, likely due to optimization instability and redundant transformations. Overall, a two-layer configuration with moderate width ($2\times$ expansion) achieves the best balance between expressivity, stability, and efficiency, and is therefore adopted as our default design.

## D.3 Ablation on the Style Extractor

To verify that the performance gain of Hystar does not rely on a specific feature extractor such as $DINOv2$, we evaluate alternative sources for deriving the style vector $z$. As shown in Table 10, using pretrained visual features (either from $DINOv2$, VGG (Simonyan & Zisserman, 2014) or the CLIP backbone itself) substantially outperforms the static baseline without external style cues. This confirms that explicit style conditioning—rather than the particular choice of extractor—is the key factor driving improvement. Moreover, the gap between $DINOv2$ and CLIP-based features is relatively small (83.6 vs. 83.0 Top-1 average), suggesting that the hypernetwork effectively adapts to diverse feature domains and that Hystar's benefit is not confounded by using another vision encoder. $DINOv2$ slightly outperforms CLIP(self), consistent with its stronger representation of texture and spatial statistics, but both yield similar cross-style generalization trends. These results validate that the proposed method's advantage arises from its adaptive mechanism, not from an unfair reliance on external feature strength.

## E Broader Generalization Studies

In this section, we extend our evaluation beyond retrieval to more diverse task settings and further include experiments on other representative vision–language representation models (VLRMs), aiming

| Extractor | Art | Sketch | Low-Res | Text | Top-1 Avg |
|---|---|---|---|---|---|
| Static Only | 65.7 | 77.0 | 83.7 | 69.2 | 73.9 |
| VGG | 73.4 | 88.5 | 96.9 | 70.1 | 82.2 |
| DINOv2 | 75.2 | 90.2 | 98.0 | 70.9 | 83.6 |
| CLIP(self) | 74.1 | 90.0 | 97.5 | 70.3 | 83.0 |

Table 10: **Ablation on style feature extractors for hypernetwork conditioning.** We compare different choices of feature sources used to derive the style vector $z$ in Hystar on the CLIP backbone across four query styles (Art, Sketch, Low-Resolution and Text). We report Top-1 accuracy (%) for each style and the average.

to systematically examine the generalization capability of the proposed hybrid modulation mechanism across different scenarios and model architectures. Specifically, we analyze its adaptability from two complementary perspectives: (1) image classification under the base-to-new transfer setting, and (2) cross-model validation on the ALBEF framework. Together, these studies provide a more comprehensive understanding of Hystar's generalization potential and the stability of the proposed method across tasks and models.

### E.1 GENERALIZATION EVALUATION ON IMAGE CLASSIFICATION

To further assess generalization, we follow the CoOp (Zhou et al., 2022c) image-classification protocol and conduct 16-shot *base-to-new* experiments on ImageNet (Deng et al., 2009) and SUN397 (Xiao et al., 2010), where each class provides 16 labeled samples for training. We compare against baselines including CoOp (Zhou et al., 2022c), ProGrad (Zhu et al., 2023a), KgCoOp (Yao et al., 2023), MaPLe (Khattak et al., 2023), and TCP (Yao et al., 2024). As summarized in Table 11, Hystar delivers the strongest generalization on *New* and *H* (harmonic mean) splits on ImageNet (New: 70.98, H: 73.93), indicating improved adaptation to unseen categories while maintaining base-domain stability. On SUN397, MaPLe attains the best *New* score (78.70), whereas TCP slightly leads on *H* (80.35); Hystar closely follows on both metrics (New: 78.41, H: 80.16), showing competitive cross-domain robustness. Averaged across datasets, Hystar achieves the best *New* (74.70) and *H* (77.03), improving over the strongest baselines (e.g., +0.08 on *New* vs. MaPLe and +0.15 on *H* vs. TCP), while accepting a marginal drop on *Base* compared to TCP (79.51 vs. 79.95). These results suggest that the proposed dynamic modulation yields a favorable trade-off between base-domain retention and out-of-domain adaptability, which is precisely the balance needed for reliable few-shot generalization.

| Datasets | Sets | CoOp | ProGrad | KgCoOp | MaPLe | TCP | Hystar(ours) |
|---|---|---|---|---|---|---|---|
| ImageNet | Base | 76.46 | 77.02 | 75.83 | 76.66 | **77.27** | 77.13 |
| | New | 66.31 | 66.66 | 69.96 | 70.54 | 69.87 | **70.98** |
| | H | 71.02 | 71.46 | 72.78 | 73.47 | 73.38 | **73.93** |
| SUN397 | Base | 80.85 | 81.26 | 80.29 | 80.82 | **82.63** | 81.89 |
| | New | 68.34 | 74.17 | 76.53 | **78.70** | 78.20 | 78.41 |
| | H | 74.07 | 77.55 | 78.36 | 79.75 | **80.35** | 80.16 |
| Average | Base | 78.66 | 79.14 | 78.06 | 78.74 | **79.95** | 79.51 |
| | New | 67.33 | 70.41 | 73.25 | 74.62 | 74.04 | **74.70** |
| | H | 72.56 | 74.52 | 75.58 | 76.62 | 76.88 | **77.03** |

Table 11: **Generalization results on 16-shot image classification benchmarks.** Each category contains 16 training samples. We evaluate cross-domain generalization from base to new classes on ImageNet and SUN397. Hystar consistently achieves the highest accuracy on *New* and *H* (harmonic mean) splits, demonstrating strong generalization to unseen categories. Best results are highlighted in **bold**.

## E.2 GENERALIZATION TO OTHER VISION-LANGUAGE REPRESENTATION MODELS (VLRMs)

To assess the generalizability of our hypernetwork-based multi-style retrieval framework beyond CLIP and BLIP, we further experiment with the ALBEF backbone. Since ALBEF consists of only 12 encoder layers, we proportionally select layers 2, 4, and 6 for hypernetwork injection. The results are summarized in Table 12. For the StyleNCE loss, we adopt the same hyperparameters as used with CLIP, without any backbone-specific tuning.

| Method | Art | Sketch | Low-Res | Text | Top-1 Avg |
|---|---|---|---|---|---|
| ALBEF | 63.7 | 52.4 | 39.1 | 61.7 | 54.2 |
| Hystar(ours) | 71.0 | 84.5 | 91.5 | 64.3 | 77.8 |

Table 12: **Multi-style retrieval performance on ALBEF backbone across four query styles** (Art, Sketch, Low-Resolution, and Text). We report Top-1 accuracy (%) for each style and the average. Our hypernetwork injection improves performance consistently, showing generalization beyond the CLIP backbone and BLIP backbone.

As shown in Table 12, the proposed hypernetwork approach improves retrieval performance across all four styles, even when applied to ALBEF without any specialized tuning. While the improvements are generally smaller than those observed on CLIP and BLIP (the main experimental backbone), this validates the cross-architecture applicability of our method. These results indicate that the benefits of middle-layer injection and cross-style feature learning are not limited to a single VLRM, supporting the generality of our approach in multi-style retrieval scenarios.

## F ANALYSIS OF SPECIAL STYLES

In this section, we further analyze Hystar's behavior under challenging and unconventional visual conditions. Specifically, we examine two aspects of style generalization beyond standard artistic domains: (1) adaptation to *extremely distinctive and unseen* styles that differ drastically from the training distribution, and (2) responses to *mixed-style queries* that blend multiple stylistic attributes within a single image. These analyses reveal how Hystar maintains semantic consistency while flexibly modeling complex or hybrid style variations.

### F.1 ANALYSIS OF EXTREMELY DISTINCTIVE STYLES

To evaluate the generalization ability of our model to extremely abstract and unseen styles, we construct an evaluation benchmark based on the DomainNet dataset. We randomly select 1,000 images from the real domain, covering 50 categories with 20 images per category, and use Stable Diffusion (Rombach et al., 2022) to generate corresponding versions in three extreme artistic styles: *Surrealist Abstract Art*, *Post-Impressionist Painting*, and *Ink-Wash Painting*. These styles are highly abstract, visually unconventional, and entirely unseen during training. We employ Stable Diffusion with the following textual prompts ( where {object} is a placeholder, e.g., "cat"):

- Surrealist Abstract Art: `A photo of a {object}, surrealist abstract art, dream-like forms, distorted proportions, fluid shapes, high contrast lighting, masterpiece.`

- Post-Impressionist Painting: `A photo of a {object}, post-impressionist oil painting, visible brush strokes, vibrant color palette, Van Gogh and Cézanne inspired, expressive texture.`

- Ink-Wash Painting: `A photo of a {object}, traditional Chinese ink-wash painting, minimal color, soft brush ink diffusion, paper texture, serene composition.`

For qualitative visualization, we display two representative samples for each style(Figure 4). Under the zero-shot setting, we test the model's ability to retrieve the correct real-domain images given

queries from these extreme styles. All models (except the original CLIP) are pretrained only on the DSR dataset. The quantitative results are reported in Table 13.

As shown in Table 13, all methods experience a noticeable performance drop under these extremely abstract and out-of-distribution styles, confirming the significant domain gap between realistic and artistic representations. Our proposed Hystar achieves the highest Top-1 accuracy across all three challenging styles, outperforming the strongest baseline by 3.5% on average. The improvements are especially pronounced on the most abstract *Surrealist Abstract Art* domain , indicating that dynamically modulated attention effectively captures style-specific variations while maintaining semantic consistency. These results highlight Hystar's superior ability to generalize across visually divergent and previously unseen artistic domains.

| Method | Surrealist Abstract Art | Post-Impressionist Painting | Ink-Wash Painting | Top-1 Avg |
|---|---|---|---|---|
| CLIP | 13.7 | 51.4 | 33.5 | 32.9 |
| LoRA | 12.4 | 47.0 | 29.4 | 29.6 |
| VPT | 16.1 | 52.6 | 31.2 | 33.3 |
| FreestyleRet | 22.8 | 70.2 | 38.6 | 43.9 |
| Hystar(ours) | **25.3** | **76.9** | **40.1** | **47.4** |

Table 13: **Retrieval performance under extreme styles.** We evaluate multiple methods on three highly distinctive style domains: *Surrealist Abstract Art*, *Post-Impressionist Painting* (Van Gogh-like), and *Ink-Wash Painting*. Results are reported as Top-1 accuracy (%) across the three extreme query types and their average. Best results are highlighted in **bold**. Our method (Hystar) achieves consistent improvements across all style types, demonstrating superior robustness and generalization to extreme and out-of-distribution visual styles.

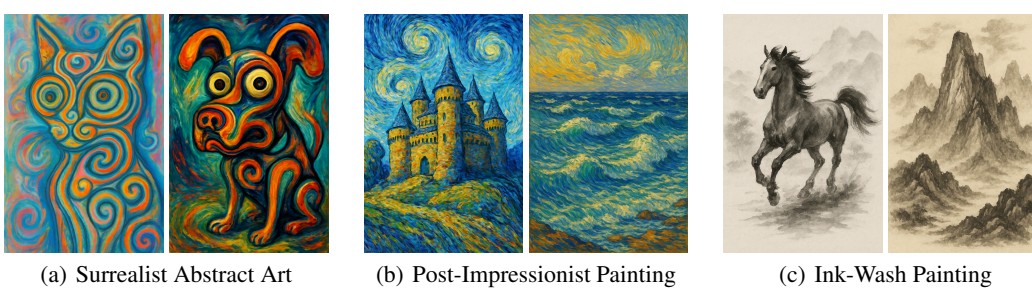

(a) Surrealist Abstract Art        (b) Post-Impressionist Painting        (c) Ink-Wash Painting

Figure 4: Examples of images with extreme styles.

## F.2 ANALYSIS OF MIXED-STYLE QUERIES

To study how Hystar responds to style mixtures, we use Stable Diffusion to synthesize three sets of images: **Sketch**, **Art**, and their **Mixture**. To control semantic content, we fix the object category and generate 50 images per set. The prompts are as follows (where {object} is a placeholder, e.g., "cat"):

- Sketch: `A {object}, clean pencil sketch, line art, monochrome, minimal shading, white background, highly detailed, professional illustration.`
- Art: `A {object}, oil painting, rich brush strokes, vibrant color palette, canvas texture, dramatic lighting, high detail, masterpiece.`
- Mixture: `A {object}, hybrid style combining oil painting and pencil sketch, partially sketched outlines, visible graphite lines with textured brush strokes, mixed-media look, coherent composition, highly detailed.`

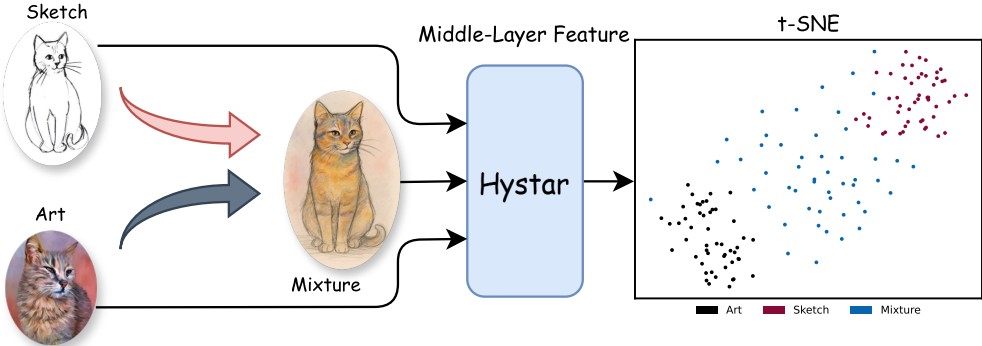

Figure 5: **Style-aware feature distribution learned by Hystar.** Hystar maps images from different styles (Art, Sketch, and their Mixture) into a coherent embedding space. As shown in the t-SNE plot, the mixed-style samples (blue) form a transitional manifold between Art (black) and Sketch (red), demonstrating that Hystar's representations smoothly capture style blending while maintaining semantic alignment.

For each image, we forward it through Hystar and extract the visual representation from a middle Transformer block of CLIP; we then visualize the features by projecting them to 2D with t-SNE. The results(Figure 5) show two compact and separable clusters for Art and Sketch, while the Mixture samples do not collapse into either cluster but instead form a continuous "bridge" between them. The bridge shifts toward the visually dominant component style, indicating that Hystar's mid-level representation varies smoothly with style strength and mixture ratio while preserving semantic consistency. This behavior suggests that Hystar encodes style in a continuous and interpretable manner and maintains good style separability under stable content representations.

## G    ADDITIONAL ABLATION OF STYLENCE LOSS

In this section, we investigate the sensitivity of the StyleNCE loss to its key hyperparameters. Specifically, we study (i) the positive–negative balance coefficient $\gamma$, which controls the relative weighting between positive and negative pairs, (ii) the hard-negative weight $\lambda$ in the OT optimization, which regulates the contribution of difficult negatives, and (iii) a comparison between StyleNCE and other loss functions. All analyses are conducted on the DSR dataset across three representative styles: Art, Sketch, and Low-Resolution.

### G.1    EFFECT OF POSITIVE-NEGATIVE BALANCE COEFFICIENT $\gamma$

Figure 6 illustrates the effect of varying $\gamma \in 1, 10, 30, 50, 80, 120, 200, 500$. When $\gamma$ is too small (e.g., $\gamma = 1$), the contribution of negative samples becomes negligible, causing training to be dominated by positives, which slows convergence and significantly degrades retrieval accuracy. Increasing $\gamma$ accelerates convergence and improves final performance, with the best results obtained in the range of $\gamma = 80$ to $\gamma = 120$. Further enlarging $\gamma$ (e.g., $\gamma = 500$) does not provide additional benefits and instead introduces slight instability, leading to performance drops relative to the mid-range values. These findings highlight the importance of maintaining a balanced contribution between positive and negative samples for stable optimization.

### G.2    SENSITIVITY ANALYSIS OF HARD-NEGATIVE WEIGHT IN OT OPTIMIZATION

Figure 7 presents a sensitivity analysis of the hard-negative weighting coefficient $\lambda \in 0.1, 0.3, 0.5, 1.0, 2.0, 3.0, 5.0, 10.0$. We observe that very small values (e.g., $\lambda = 0.1$) place excessive emphasis on hard negatives, causing the model to largely ignore easy negatives, which destabilizes training and reduces performance. Conversely, excessively large values (e.g., $\lambda = 10.0$) un-

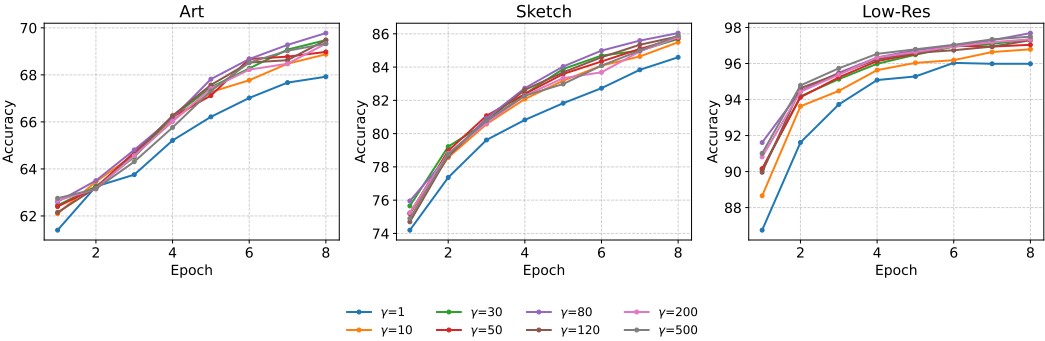

Figure 6: Effect of the positive–negative balance coefficient $\gamma$ on DSR retrieval accuracy(%). Results are reported for three styles: Art, Sketch, and Low-Resolution. Small values of $\gamma$ (e.g., 1) result in slower convergence and lower accuracy, while moderate values ($80 \leq \gamma \leq 120$) achieve the best trade-off between stability and performance.

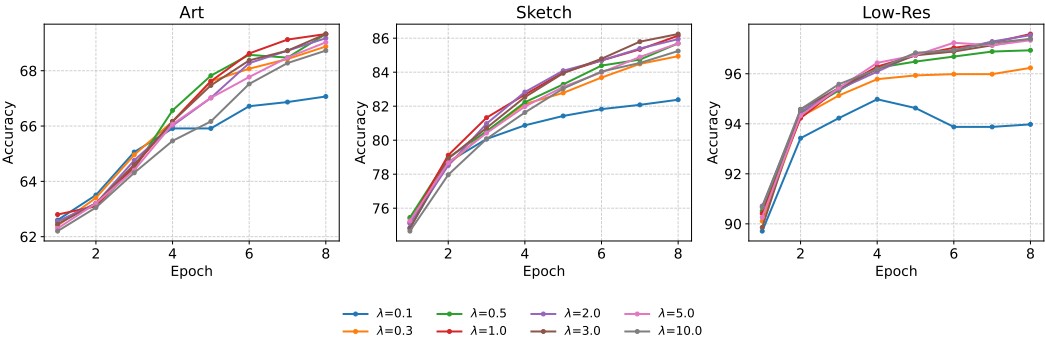

Figure 7: Sensitivity analysis of the hard-negative weight $\lambda$ in OT optimization on DSR retrieval accuracy(%). Results are reported for Art, Sketch, and Low-Resolution. Moderate values ($1.0 \leq \lambda \leq 3.0$) yield the best performance. Very small values underweight easy negatives, whereas very large values fail to effectively exploit hard negatives, limiting performance.

derweight hard negatives, resulting in insufficient hard-negative mining and moderate performance degradation. Consistently high retrieval accuracy and stable convergence are observed for intermediate values ($\lambda = 1.0$–$3.0$). These findings indicate that appropriately balancing the contribution of hard negatives is critical for fully leveraging OT-based optimization.

### G.3 COMPARISON OF STYLENCE AND OTHER LOSS FUNCTIONS

| Method | Art | Sketch | Low-Res | Top-1 Avg |
|---|---|---|---|---|
| Triplet loss | 65.6 | 71.9 | 89.5 | 75.7 |
| InfoNCE loss | 70.2 | 85.3 | 94.2 | 83.2 |
| Circle loss | 70.4 | 88.8 | 96.1 | 85.1 |
| Triplet loss + Hard Negative Sampling | 69.3 | 80.2 | 93.0 | 80.8 |
| InfoNCE loss + Hard Negative Sampling | 72.6 | 88.4 | 96.7 | 85.9 |
| StyleNCE loss(Ours) | **75.2** | **90.2** | **98.0** | **87.8** |

Table 14: **Ablation study of different loss functions on the CLIP backbone.** Results are reported as Top-1 accuracy (%) across the three major query types used for training (Art, Sketch, and Low-Resolution) and their average. Best results are highlighted in **bold**.

From Table 14, we observe that the choice of loss function has a significant impact on retrieval performance. The baseline Triplet loss achieves the lowest accuracy (75.7% on average), indicating its limited ability to handle cross-style variation. InfoNCE and Circle loss provide clear improvements (83.2% and 85.1%), thanks to their better optimization of inter-class separation. Incorporating hard negative sampling further boosts performance for both Triplet and InfoNCE, but the gain is relatively modest (+5.1 and +2.7 points, respectively), suggesting that negative mining alone cannot fully address style discrepancies. In contrast, our proposed StyleNCE achieves the best results across all three query types, surpassing the best baseline (InfoNCE + hard negatives) by +1.9 points on average. This demonstrates that StyleNCE not only benefits from hard negative mining but also explicitly models style-aware feature alignment, leading to consistent gains across diverse query styles.

## H    RETRIEVAL RESULT VISUALIZATION

### H.1    RETRIEVAL RESULT VISUALIZATION ON DSR

To provide qualitative insights into model behavior, we visualize retrieval examples on the **DSR** dataset (Figure 8). Typical errors are categorized into three groups: (a) *action errors*, where retrieved samples contain the correct object but incorrect actions; (b) *object errors*, where retrieved images contain semantically related but incorrect objects; and (c) *background errors*, where retrievals confuse similar contexts while missing the correct foreground. As shown in Figure 8, baseline methods frequently suffer from these mistakes, returning visually similar but semantically wrong samples. In contrast, our proposed Hystar consistently retrieves semantically accurate images across different styles, demonstrating its ability to align fine-grained semantics under challenging cross-style conditions better.

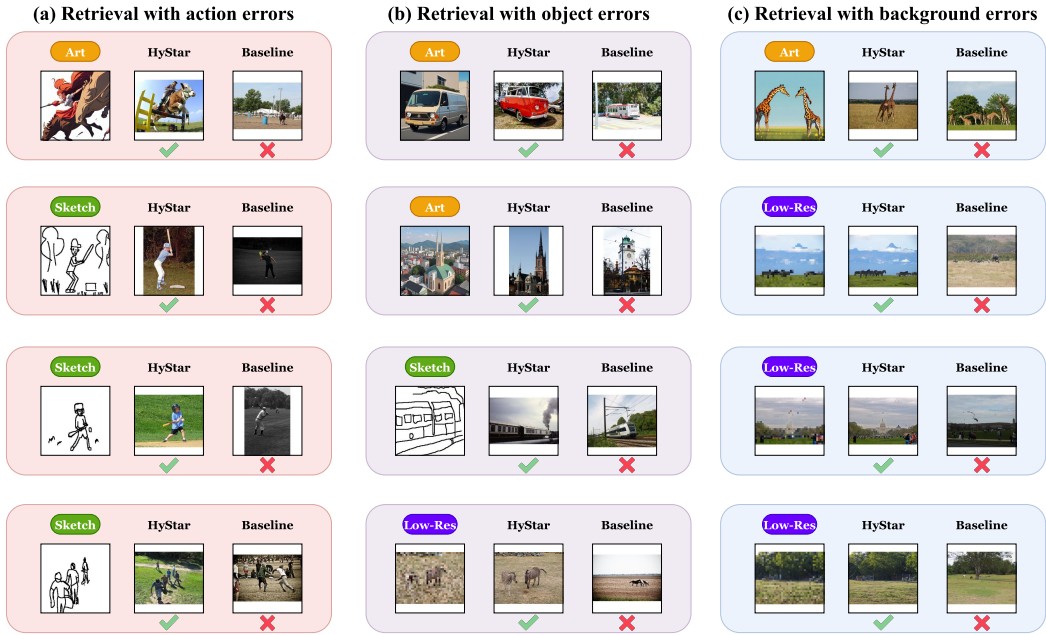

Figure 8: Qualitative retrieval examples on the DSR dataset. We illustrate three common error types made by baseline methods: (a) action errors, (b) object errors, and (c) background errors. Baselines often retrieve visually similar but semantically incorrect results, while our Hystar consistently retrieves the correct matches, highlighting its superior fine-grained alignment across multiple styles.

### H.2    RETRIEVAL RESULT VISUALIZATION ON DOMAINNET

We further evaluate retrieval results on the more diverse **DomainNet** dataset, which contains unseen styles such as Clipart, Sketch, Painting, Quickdraw, and Infograph. Figure 9 10 11 show Top-10

retrieval examples. Baseline methods often fail under large style shifts, retrieving visually close but semantically irrelevant samples, especially in abstract styles such as Quickdraw and Infograph. In contrast, our Hystar maintains stable cross-style alignment, retrieving semantically correct results across multiple unseen styles. These results demonstrate the strong generalization ability of Hystar beyond the training distribution, confirming its robustness under zero-shot cross-style retrieval.

**Clipart**

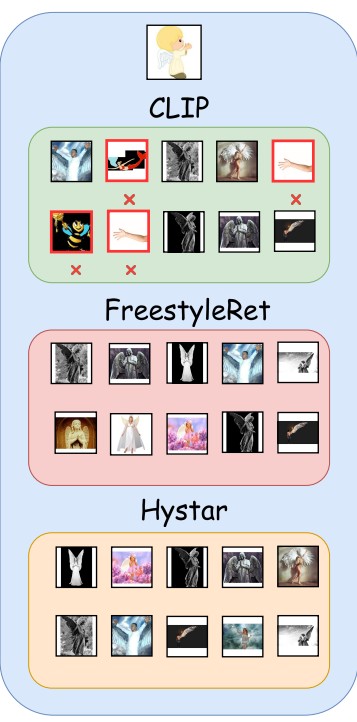 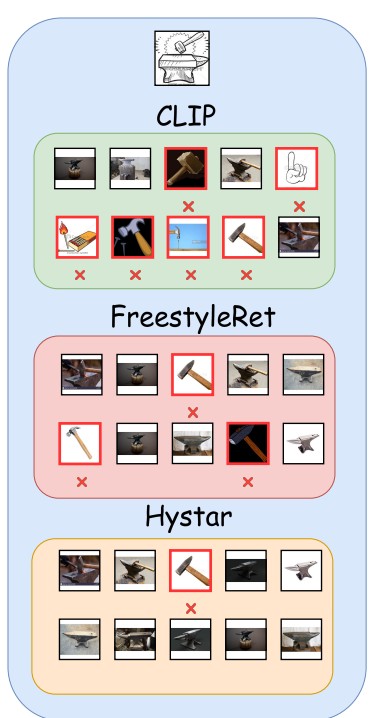

**Sketch**

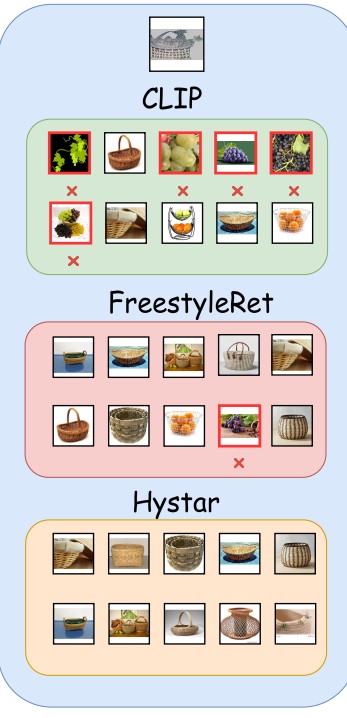 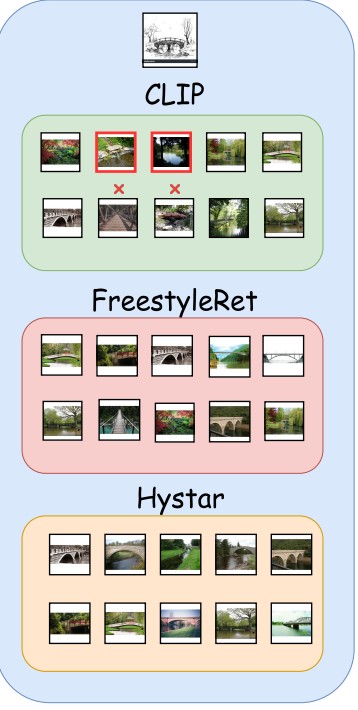

Figure 9: Qualitative Top-10 retrieval results on the DomainNet dataset across unseen styles (Clipart, Sketch). In the retrieval results figure, we use the retrieval outputs from the baseline models, CLIP and FreestyleRet, as our baseline comparison.

**Painting**

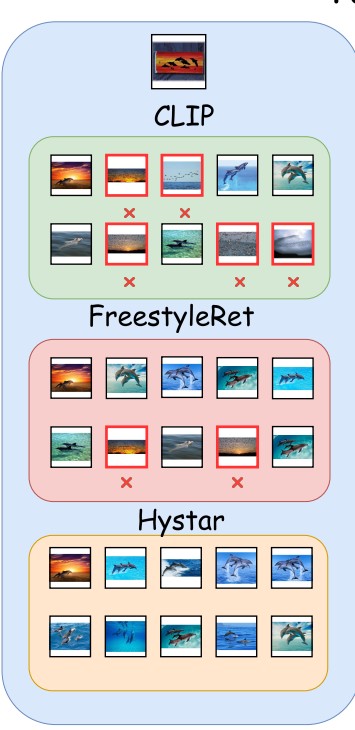 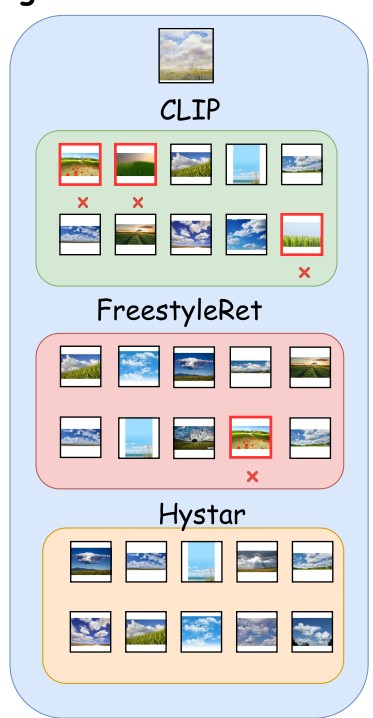

**Quickdraw**

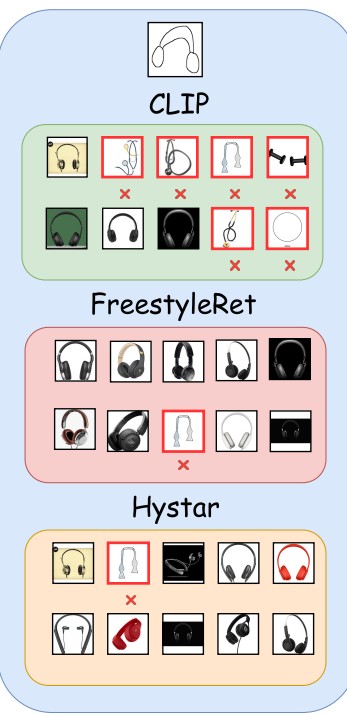 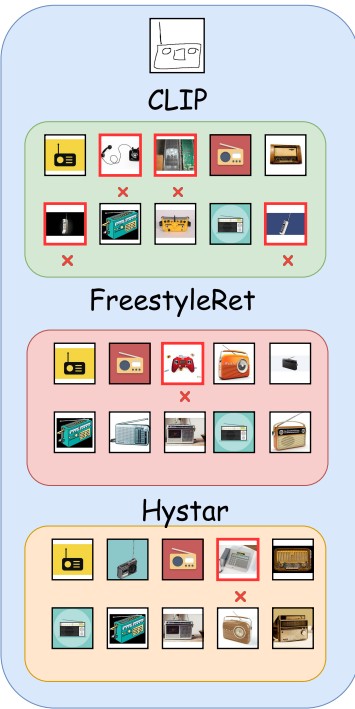

Figure 10: Qualitative Top-10 retrieval results on the DomainNet dataset across unseen styles (Painting, Quickdraw). In the retrieval results figure, we use the retrieval outputs from the baseline models, CLIP and FreestyleRet, as our baseline comparison.

Figure 11: Qualitative Top-10 retrieval results on the DomainNet dataset across unseen styles (Infograph). In the retrieval results figure, we use the retrieval outputs from the baseline models, CLIP and FreestyleRet, as our baseline comparison.

