# OpenReview forum: "Hystar: Hypernetwork-driven Style-adaptive Retrieval via Dynamic SVD Modulation"
_ICLR.cc/2026/Conference — ICLR 2026 Poster_

### Official Review · Reviewer_eVsi · 2025-10-28

**Soundness:** 4
**Presentation:** 3
**Contribution:** 3
**Rating:** 8
**Confidence:** 5

**Summary:**

This paper introduces Hystar, a lightweight framework for Query-based Image Retrieval (QBIR) that addresses the challenge of handling stylistically diverse queries (e.g., sketches, artworks) which cause distribution shifts and performance drops in standard Vision-Language Models (VLMs) like CLIP. Extensive experiments show that Hystar achieves state-of-the-art performance on multi-style retrieval and cross-style classification tasks, offering superior parameter efficiency and stable performance across various query styles.

**Strengths:**

1. Well-Motivated Problem and Thorough Literature Review: The paper adeptly identifies and tackles the critical, yet underexplored, challenge of style-diversified retrieval in QBIR. The authors provide a comprehensive overview of existing paradigms, such as LoRA-based PEFT methods and those relying on style-cluster priors, and convincingly argue their limitations in generalizing to unseen query styles. This strong motivation establishes a clear and valuable niche for their work.

 2. Innovative and Principled Methodology: The core technical contribution is both novel and well-designed. The use of a hypernetwork to dynamically generate LoRA matrices is a clever adaptation. More importantly, the decision to restrict the hypernetwork's output to singular-value perturbations (∆S) is a key insight. This approach not only enhances training stability but also explicitly promotes generalization across diverse and unseen styles by focusing adaptation on a compact, semantically meaningful parameter space.

 3. Compelling and Multi-faceted Empirical Validation: The authors provide rigorous experimental evidence to support their claims. Through comprehensive comparisons against strong baselines (e.g., FreestyleRet, VPT), they convincingly demonstrate the superiority of their hypernetwork-driven paradigm. Furthermore, the inclusion of t-SNE visualizations offers an intuitive and powerful qualitative analysis, effectively illustrating how Hystar learns more discriminative and style-invariant feature representations.

**Weaknesses:**

1. Terminological Imprecision: The term "Vision-Language Models (VLMs)" is used to describe CLIP. However, in contemporary literature, "VLM" often refers more specifically to models that include a text decoder for generative tasks (e.g., GPT-4V, LLaVA). CLIP, being a dual-encoder model designed primarily for contrastive learning and representation embedding, is more accurately categorized as a vision-language representation model. Adopting this more precise terminology would enhance conceptual clarity.

 2. Insufficient Justification for Architectural Choices: The rationale behind using a hypernetwork to generate dynamic components for attention layers while using static offsets for MLP layers, as a specific form of decoupling, requires deeper analysis. The paper would benefit from a more thorough discussion comparing this design choice against other potential decoupling strategies. A theoretical or empirical justification for why this particular split (dynamic attention vs. static MLP) is optimal for the task of style adaptation would significantly strengthen the methodological argument.

 3. Limited Scope of Ablation Studies: The ablation studies, while validating the proposed components, could be more comprehensive by including a wider range of Parameter-Efficient Fine-Tuning (PEFT) baselines. For instance, comparing against other adaptive tuning methods like (IA)³ or Adapter-based approaches would provide a clearer picture of Hystar's performance gains relative to the broader landscape of efficient adaptation techniques, not just the ones it directly outperforms.

**Questions:**

please see above

---

> ### Author Response · Authors · 2025-11-23
> **Response to eVsi**
>
> We sincerely thank the reviewer for the positive and encouraging feedback, and for recognizing the strong motivation, principled design, and comprehensive empirical validation of our work.We address each point below.
>
> **List of changes in the manuscript:**
>
> >1. **Entire manuscript:** Revised the terminology throughout the entire manuscript, replacing **“vision-language  model(VLM)”** with the more precise term **“vision-language representation model (VLRM)”** for consistency and conceptual accuracy..
> >2. **Appendix §B:** Added analysis for **dynamic attention and static MLP design**.
> >3. **Tab 1:** Added **experimental results for additional PEFT baselines.**
>
> > W1: Use more precise and formal terminology.
>
> **A:** We thank the reviewer for this helpful clarification regarding terminology.
>  We agree that “Vision-Language Representation Model (VLRM)” is a more precise term for CLIP, as it better reflects its dual-encoder and contrastive-learning nature rather than a generative framework.
>  Accordingly, we have revised the manuscript to consistently use **“vision-language representation model (VLRM)”** in place of “**vision-language model(VLM)**” to improve conceptual clarity and alignment with contemporary literature.
>
> > W2: Provide  analyses on the decoupling between dynamic and static modulation.
>
> **A:** We thank the reviewer for the insightful question regarding the rationale of employing **dynamic modulation in attention layers** and **static offsets in MLP layers** as a specific form of decoupling.
>  To address this concern, we have added a dedicated section titled **“Analysis for Dynamic Attention and Static MLP Design”** (see **Appendix §B**) .
>
> Empirically, we compare three variants :
>  (1) **Original design** (dynamic attention + static MLP),
>  (2) **Reversed design** (dynamic MLP + static attention), and
>  (3) **Dynamic-All design** (both attention and MLP dynamically modulated).
>
> | **Method**          | **Art**  | **Sketch** | **Low-Res** | **Text** |
> | ------------------- | -------- | ---------- | ----------- | -------- |
> | Dynamic-All Design  | 63.7     | 69.9       | 74.8        | 66.3     |
> | Reversed Design     | 64.1     | 71.2       | 80.2        | 69.1     |
> | **Original Design** | **75.2** | **90.2**   | **98.0**    | **70.9** |
>
>  As shown in the table, the **original design consistently achieves the highest Top-1 accuracy** across all style domains. We have added this comparison **in Appendix B as Table 6.**
>  The **Dynamic-All** variant performs the worst, as excessive flexibility destabilizes training, while the **Reversed** variant underperforms, suggesting that MLP modulation is less effective for modeling style variations.
>
> Conceptually, attention layers govern cross-token relationships that are directly influenced by style cues such as texture and composition, making them well-suited for dynamic modulation.
>  In contrast, MLP layers primarily refine semantic features within tokens; keeping them static preserves semantic stability and prevents overfitting to transient style attributes.
>  Together, these analyses confirm that our **hybrid decoupling strategy—dynamic attention for style adaptation and static MLPs for semantic consistency—achieves the optimal balance between adaptability and stability**, as now discussed in the revised manuscript ( **Appendix B , Table 6** ).
>
> > W3: Provide more PEFT approaches as baselines for a more comprehensive comparison
>
> **A:** We thank the reviewer for this helpful suggestion to broaden the ablation scope with additional PEFT baselines.
>  In response, we have **expanded the main comparison table** to include more representative parameter-efficient adaptation methods, such as **(IA)$^3$** , **Adapter-based approaches(AdaptFormer)** , **SSF** , in addition to LoRA, VPT, and FreestyleRet.
>
> The results of the newly added baselines are presented in the table below.
>
> | Method      |   **Art**   |  **Sketch**  | **Low-Res** |  **Text**   |
> | :---------- | :---------: | :----------: | :---------: | :---------: |
> |             | Top-1/Top-5 | Top-1/Top-5  | Top-1/Top-5 | Top-1/Top-5 |
> | (IA)$^3$    | 64.3 / 96.8 | 71.8 / 95.7  | 80.9 / 96.1 | 70.1 / 96.6 |
> | AdaptFormer | 65.1 / 97.0 | 73.5 / 96.4  | 81.1 / 96.3 | 69.7 / 95.8 |
> | SSF         | 64.7 / 96.4 | 73.0 /  97.0 | 79.9 / 95.8 | 70.1 / 96.3 |
>
>  This comprehensive comparison provides a clearer picture of Hystar’s performance within the broader PEFT landscape.
>  The results show that Hystar consistently achieves superior performance while maintaining competitive parameter efficiency, further validating the effectiveness of our dynamic modulation framework.
>
> We have added the results in the **main text (Table 1)**.
>
>
>
> **Conclusion:**  We sincerely thank the reviewer for the constructive feedback, which has substantially improved the clarity and rigor of our work.

---

### Official Review · Reviewer_RvYN · 2025-10-31

**Soundness:** 3
**Presentation:** 2
**Contribution:** 2
**Rating:** 6
**Confidence:** 4

**Summary:**

This work addresses the style distribution shift problem in query-based image retrieval (QBIR) (such as queries with styles different from those in the training data, like sketches and artworks), and proposes the Hystar framework. The Hystar framework includes two main components: 1. Hypernetwork-driven Dynamic SVD Modulation: A hypernetwork generates singular value perturbations (ΔS) in the attention layer based on the query style, achieving dynamic adaptation for each input. Simultaneously, static singular value offsets are used in the MLP layer to ensure cross-style stability.
2. StyleNCE Loss Function: A contrastive loss weighted by optimal transport theory highlights indistinguishable cross-style negative samples, improving the model's robustness to style differences.

**Strengths:**

1. The motivation of this work is good, which clearly points out the performance degradation problem of VLM under style distribution shift and emphasizes the static limitations of existing PEFT methods (such as LoRA and VPT).
2. The experiment was comprehensive and convincing.
3. The proposed method is efficient in parameters: Only a few parameters need to be fine-tuned.

**Weaknesses:**

1. This work emphasizes parameter efficiency, but lacks a quantitative analysis of the forward computational overhead of the hypernetwork (relative to static PEFT) (e.g., the percentage increase in inference time). This is important for evaluating its feasibility in real-time retrieval scenarios.
2. Appendix A provides an intuitive theoretical explanation of SVD modulation (based on spectral norm constraints). However, further exploration is possible, for example, analyzing from the perspectives of generalization theory or manifold learning why SVD modulation is particularly effective for style adaptation.
3. This work mainly focus on retrieval and zero-shot classification. Further validation of Hystar's generalization ability on other style-sensitive tasks can be achieved, such as stylized image generation guidance and cross-style visual question answering.

**Questions:**

1. This paper chose DINOv2 as the style feature extractor. Were other features considered (e.g., features from dedicated style analysis networks)? Do different feature choices significantly impact performance?
2. How does Hystar handle styles that are completely new, extremely abstract, or bizarre in the training data? How does the model respond when the same query image contains a mixture of multiple styles?

---

> ### Author Response · Authors · 2025-11-23
> **Response to RvYN (1/4)**
>
> We sincerely thank the reviewer for recognizing the motivation, comprehensive experiments, and parameter efficiency of our work. Below, we address each of your comments point by point.
>
> **List of changes in the manuscript:**
>
> >1. **Line 519 - 527 , Tab 5 :** Added an analysis of **parameter efficiency and inference latency**.
> >2. **Line 199 - 215:** Added an analysis from a **manifold perspective** explaining why SVD modulation is particularly effective for style adaptation.
> >3. **Appendix §E.2 , Tab 12 ,  Figure  4 :** Added experiments on the generalization of Hystar to stylized image generation guidance.
> >4. **Appendix §D.3 ,  Tab 10 :**  Added ablation on the **style extractor**.
> >5. **Appendix §F ,Tab 14 , Figure 5 , Figure 6 :** Added analysis of **special styles  and mixed styles.**
>
> > W1:  Provide parameter efficiency analysis and inference latency evaluation
>
> **A:** We thank the reviewer for the suggestion to include a more detailed analysis of parameter efficiency and inference cost.
>  In response, we have added a **computation comparison**   that quantitatively compares the model size and inference latency of Hystar with representative CLIP-based PEFT baselines including VPT, (IA)$^3$, AdaptFormer, and FreestyleRet.
>
> | **Method**   | **Parameters (M)** | **Additional Params (%)** | **Speed (ms)** | **Inference Time Increase (%)** |
> | ------------ | ------------------ | ------------------------- | -------------- | ------------------------------- |
> | CLIP         | 427M               | —                         | 68             | —                               |
> | VPT          | 428M               | 0.2                       | 73             | 7.4                             |
> | (IA)³        | 427M               | 0.1                       | 71             | 2.9                             |
> | AdaptFormer  | 429M               | 0.5                       | 74             | 8.8                             |
> | FreestyleRet | 476M               | 11.5                      | 96             | 41.2                            |
> | Hystar       | 442M               | 3.5                       | 108            | 58.8                            |
>
> The results show that **Hystar maintains competitive efficiency**, with only a modest 3.5 % increase in parameters and 58.8 % inference-time overhead relative to CLIP, while achieving substantially better robustness to style and domain variations.
>  This additional cost mainly comes from the controller-guided hypernetwork module, which dynamically modulates representations for style adaptation.
>  Overall, the results demonstrate that **Hystar achieves a good balance between adaptability and efficiency**, remaining compact and practical for real-time retrieval applications.
>
> We have added this comparison in **Line 519 - 527 , Tab 5**.
>
> > W2: Provid theoretical intuition from a manifold learning perspective to explain why SVD-based modulation is effective for style adaptation.
>
> **A:** We thank the reviewer for the insightful suggestion to strengthen the theoretical discussion of SVD modulation.
>  We have discussed the stability of SVD modulation in Appendix A. Furthermore, we have added a theoretical explanation  to clarify its foundation from a manifold learning perspective.
>
> Specifically, we now highlight that modulating singular values offers not only **stability** but also a **geometric interpretation** aligned with style adaptation.
>  In our formulation, $U$ and $V$ define the pretrained semantic subspace, while the singular values $\Sigma$ determine the scaling along these spectral directions.
>  By modulating only the singular values while keeping $U$ and $V$ fixed, the model performs **geometry-preserving deformations** within the pretrained manifold, smoothly rescaling existing semantic directions according to style-induced variations.
>  This explains why SVD modulation achieves flexible yet stable adaptation across styles.
>
> In contrast to LoRA, which introduces new low-rank directions that may distort pretrained semantics, SVD modulation confines updates within the **original spectral subspace**, maintaining semantic consistency while adjusting feature intensity.
>  This spectral alignment provides a natural manifold-based interpretation of why SVD modulation generalizes better under style shifts.
>  We have incorporated this theoretical rationale into the revised manuscript  **(Line 199 - 215)** to complement the existing spectral-norm analysis .

---

> ### Author Response · Authors · 2025-11-23
> **Response to RvYN (2/4)**
>
> > W3: Provid additional experiments to demonstrate the effectiveness of our framework on broader style adaptation tasks.
>
> **A:**  We thank the reviewer for the valuable suggestion to further validate **Hystar’s generalization ability** beyond retrieval and zero-shot classification.
>  In response, we have conducted **additional experiments on stylized image generation guidance**, following the few-shot generation protocols proposed in **AdAM and RICK**.
>
> Specifically, we adapt Hystar to the **StyleGAN2** backbone using a **selective hybrid modulation scheme**:
>  dynamic modulation is applied only to the convolutional layers most sensitive to style variations,
>  while static SVD-based modulation is used for the remaining layers to ensure semantic stability and low-rank regularization.
>  The dynamic parameters are generated by a lightweight hypernetwork conditioned on the latent code (w) from the StyleGAN2 mapping network.
>
> | **Method**        | **Sketches** | **AFHQ-Cat** | **FID-Avg** |
> | ----------------- | ------------ | ------------ | ----------- |
> | TGAN              | 53.42        | 64.68        | 59.05       |
> | TGAN+ADA          | 66.99        | 80.16        | 73.58       |
> | FreezeD           | 46.54        | 63.60        | 55.07       |
> | EWC               | 64.55        | 74.61        | 69.58       |
> | CDC               | 47.62        | 176.21       | 111.92      |
> | RSSA              | 69.51        | 159.54       | 114.53      |
> | SoLAD             | 37.23        | 61.35        | 49.29       |
> | AdAM              | 42.64        | 58.07        | 50.36       |
> | RICK              | 35.66        | **53.27**    | 44.47       |
> | **Hystar (Ours)** | **34.12**    | 54.68        | **44.40**   |
>
> As shown in  Table , Hystar achieves the **lowest average FID** across two stylized domains (**Sketches** and **AFHQ-Cat**) and outperforms all compared baselines (**TGAN, TGAN+ADA ,FreezeD, EWC, CDC, RSSA, SoLAD, AdAM, RICK**).
>
> We have added this comparison in **Appendix E.2 as Table 12**.
>
>  These results confirm that **Hystar’s dynamic spectral modulation** effectively captures domain-specific style cues while maintaining semantic consistency.
>  Overall, the experiment demonstrates that Hystar **generalizes well to other style-sensitive tasks**, highlighting its potential applicability to broader domains such as stylized generation and cross-style reasoning.  We have updated the manuscript accordingly **(Appendix E.2 , Table 12)**.
>
> The proposed framework can also be extended to broader tasks such as multi-style visual question answering and other  applications. We plan to explore and validate these directions in future work, aiming to develop a more unified and general framework.

---

> ### Author Response · Authors · 2025-11-23
> **Response to RvYN (3/4)**
>
> > Q1:  More ablation study and experimental comparisons about DINOv2
>
> **A:** We thank the reviewer for raising this question regarding the choice of the style feature extractor.
>  To verify that Hystar’s performance improvement does not depend on a specific feature extractor such as **DINOv2**, we conducted an **ablation study on the style extractor**  .
>
> Here is the ablation on style feature extractors for hypernetwork conditioning (We have added this comparison in **Appendix D.3 as Table 10**.) ：
> | **Extractor** | **Art** | **Sketch** | **Low-Res** | **Text** | **Top-1 Avg** |
> | :------------ | :-----: | :--------: | :---------: | :------: | :-----------: |
> | Static Only   |  65.7   |    77.0    |    83.7     |   69.2   |     73.9      |
> | VGG           |  73.4   |    88.5    |    96.9     |   70.1   |     82.2      |
> | DINOv2        |  75.2   |    90.2    |    98.0     |   70.9   |     83.6      |
> | CLIP(self)    |  74.1   |    90.0    |    97.5     |   70.3   |     83.0      |
>
> We compared multiple sources for deriving the style vector $z$, including **DINOv2**, **VGG**, and the **CLIP backbone itself**.
>  As shown in the results, all pretrained visual features (DINOv2, VGG, or CLIP) substantially outperform the static baseline without explicit style conditioning, confirming that **the key factor is the adaptive style-conditioning mechanism itself**, rather than reliance on a particular extractor.
>  Moreover, the gap between DINOv2 and CLIP-based features is small (83.6 vs. 83.0 Top-1 Avg), indicating that Hystar generalizes well across different feature domains and is not biased by the use of an additional vision encoder.
>  DINOv2 achieves a slightly higher score, consistent with its stronger texture and spatial representation capacity, but the overall trend remains consistent across all extractors.
>
> We have included this analysis in the revised manuscript (**Appendix D.3 , Table 10**)  to clarify that Hystar’s advantages arise from **its dynamic adaptive mechanism**.

---

> ### Author Response · Authors · 2025-11-23
> **Response to RvYN (4/4)**
>
> > Q2: Provided additional evaluations of our framework on extremely distinctive styles and included an analysis of its responses to mixed-style inputs.
>
> **A:** We thank the reviewer for the valuable question regarding Hystar’s behavior under unseen, highly abstract, and mixed-style visual conditions.
>  To address this, we have added a new section titled **“Analysis of Special Styles”** (see **Appendix §F** ), which includes two complementary experiments:
>
> **(1) Extremely distinctive and unseen styles.**
>  We construct an evaluation benchmark based on the **DomainNet** dataset, where 1,000 real-domain images are translated into three extreme artistic styles, namely **Surrealist Abstract Art**, **Post-Impressionist Painting**, and **Ink-Wash Painting**, using Stable Diffusion.
>
> | **Method**        | **Surrealist Abstract Art** | **Post-Impressionist Painting** | **Ink-Wash Painting** | **Top-1 Avg** |
> | ----------------- | --------------------------- | ------------------------------- | --------------------- | ------------- |
> | CLIP              | 13.7                        | 51.4                            | 33.5                  | 32.9          |
> | LoRA              | 12.4                        | 47.0                            | 29.4                  | 29.6          |
> | VPT               | 16.1                        | 52.6                            | 31.2                  | 33.3          |
> | FreestyleRet      | 22.8                        | 70.2                            | 38.6                  | 43.9          |
> | **Hystar (Ours)** | **25.3**                    | **76.9**                        | **40.1**              | **47.4**      |
>
>  As shown in the Table , all methods show a noticeable performance drop under these highly abstract, out-of-distribution styles, but **Hystar achieves the highest Top-1 accuracy across all three**, outperforming the strongest baseline by **3.5 % on average**.
>  This demonstrates that dynamically modulated attention allows Hystar to flexibly adapt to unseen style distributions while preserving semantic alignment.
>
> We have added this table in **Appendix F.1 as Table 14.**
>
>  Qualitative examples of generated styles are provided in **Appendix F.1 as Figure 5**.
>
> **(2) Mixed-style queries.**
>  To analyze Hystar’s response to hybrid styles, we synthesize **Sketch**, **Art**, and **Mixture** images using Stable Diffusion.
>  We visualize the resulting feature embeddings with **t-SNE** (see **Appendix F.2 , Figure 6**).
>  The embeddings of mixed-style samples form a continuous “bridge” between the two distinct clusters of Sketch and Art, indicating that Hystar encodes styles in a **continuous and interpretable manifold**, where representations vary smoothly with the mixture ratio and maintain semantic consistency.
>
> Together, these analyses confirm that **Hystar generalizes robustly to completely new and highly abstract styles**, and that its representations capture **style blending in a smooth, geometry-aware manner**, preserving stable semantics even under complex or hybrid visual conditions.
>
>
>
> **Conclusion:**  We sincerely thank the reviewer for the constructive feedback, which has substantially improved the clarity and rigor of our work.

---

### Official Review · Reviewer_vPC8 · 2025-11-01

**Soundness:** 2
**Presentation:** 3
**Contribution:** 2
**Rating:** 4
**Confidence:** 4

**Summary:**

This paper introduces HySTAR, a novel framework for query-based image retrieval (QBIR) designed to handle significant variations in query style (e.g., sketches, artwork, low-resolution images). The core problem it addresses is the poor performance of large vision-language models (VLMs) like CLIP when faced with stylistic distribution shifts.

The paper's two main contributions are:
- Hypernetwork-driven Dynamic PEFT: A parameter-efficient fine-tuning (PEFT) method where a small hypernetwork dynamically generates input-specific modulations. These modulations are applied to the singular values (SVD) of the attention layers in a frozen VLM, conditioned on a style feature vector.
- StyleNCE Loss: A contrastive loss function that uses optimal transport (OT) to re-weight hard negative samples, specifically designed to improve semantic alignment across different styles- first to do so for Image Retrieval.

The authors claim that this combination of dynamic, style-adaptive modulation and a style-focused loss function achieves state-of-the-art results on the DSR and DomainNet benchmarks.

**Strengths:**

- Relevant Problem: The paper tackles a well-known and practical limitation of large pre-trained models: their lack of robustness to domain and style shifts. The application to cross-style image retrieval is a challenging and valuable research area.
 - Novel Methodology: The primary technical idea—using a hypernetwork to predict dynamic PEFT parameters based on the input's style—is a creative and novel contribution. Moving from static PEFT (like LoRA or VPT) to an adaptive, input-conditioned PEFT is an interesting research direction. The specific choice to modulate singular values (SVD) is also an unconventional and intriguing approach.
 - Well-Motivated Loss Function: The proposed StyleNCE loss is a strong contribution. The intuition that cross-style retrieval creates many "hard negatives" (e.g., a sketch of a cat vs. a photo of a tiger) is sharp. Using optimal transport to systematically identify and up-weight these hard samples is a well-founded and logical approach to improving fine-grained, cross-style discriminative power.

**Weaknesses:**

Despite the novel ideas, the paper suffers from several major weaknesses, primarily in the experimental design and the clarity of the method's description.
- Unfair Experimental Comparison (Major Flaw): The central claim of HySTAR's superiority is built on a confounded experiment. As shown in Figure 1, HySTAR's hypernetwork is conditioned on features from DINOv2, a powerful, separate vision model. The baseline methods (CLIP, BLIP, LoRA, VPT, FreestyleRet) do not have access to these supplementary features. The performance gains attributed to HySTAR's architecture (dynamic SVD modulation) may, in large part, be coming from the simple fact that it is a multi-model system using rich DINOv2 features that the baselines lack. This is a critical confounding variable that makes the primary results in Tables 1, 2, and 3 uninterpretable.
- Missing Architectural and Implementation Details: The paper is missing key details required for understanding and reproducing the method.
  + Hypernetwork Architecture: The hypernetwork is defined in Eq. 2 as a simple 2-layer MLP, but its dimensions (input, hidden, and output) are never specified.
  + Style Feature Extraction: It is unclear how the style feature $z$ is extracted from DINOv2. Is it the [CLS] token of the input image? An average pool of patch tokens? This is a crucial, undefined component of the model.
- Insufficient Ablation Studies: The ablation study in Table 4, while showing the benefit of the "Hyper" and "Static" modules, is insufficient. There are no ablations on the hypernetwork design itself (e.g., depth, width) to justify the chosen (and unstated) architecture.
- Limited Theoretical Justification for SVD: The paper's justification for using SVD modulation is thin. Appendix A argues that it provides stability by bounding the spectral norm of the weight update. However, it provides no theoretical or empirical intuition as to why modulating singular values is semantically the correct approach for style adaptation. Why is this superior to other low-rank updates like LoRA ($W_0 + BA$), especially when LoRA is more computationally straightforward (it doesn't require SVD on-the-fly)? The connection between "visual style" and "singular values of attention weights" is never established.
- Errors and Poor Presentation in Results:
   + Error in Table 2: As noted by the user, the paper incorrectly highlights its own score as "best" in Table 2. For the "Infograph" style (last column), the Top 1 accuracy for CLIP is 60.3%, which is higher than HySTAR's 59.3%. Yet, HySTAR’s score is bolded.
    + Weak Visualization (Figure 7): The figures are extremely hard to read. It is also one of the most prominent results showcasing the comparison between baseline method and your method for retrieval across various styles. Also, it is not clear what is the baselines comparison? Is it zero shot CLIP? Or FreestyleRet? FreestyleRet has better retrieval results if so, than what is demonstrated in the comparison.

**Questions:**

- Weakness 1): Can you please address the experimental confound of using DINOv2? To fairly evaluate HySTAR as a PEFT method, could you provide an ablation comparing the baselines (VPT, LoRA) to a version of HySTAR that does not use DINOv2, and instead derives its style vector $z$ from the VLM backbone itself (e.g., from CLIP)?
- Re: (Weakness 2): What are the precise architectural details of the hypernetwork (input, hidden, and output dimensions)? And how exactly is the style feature $z$ extracted from the DINOv2 model?
- Re: (Weakness 3): Beyond stability, what is the theoretical or empirical justification for SVD modulation being a superior method for style adaptation compared to other low-rank updates like LoRA?
- Re: (Weakness 4 / Figure 7): Please clarify what the baselines are and make the figures readable to verify the claims of superior consistency.

---

> ### Author Response · Authors · 2025-11-23
> **Response to vPC8 (1/4)**
>
> Thank you for the thoughtful review and for recognizing the significance of domain/style shift and the novelty of our hypernetwork-based dynamic PEFT and StyleNCE loss. We address each point below.
>
> **List of changes in the manuscript:**
>
> >1. **Appendix §D.3 , Tab 10:** Added ablation on the **style extractor**.
> >2. **Line 231 - 235:** Added clarifications on the **dimensionality of the hypernetwork**.
> >3. **Line 223 - 226:** Added details on the process of **extracting style representations from DINOv2**.
> >4. **Appendix §D.2 , Tab 9 :**  Added ablation studies on the **width and depth of the hypernetwork**.
> >5. **Appendix §B ,  Tab 6 :** Added analysis for **dynamic attention and static MLP design**.
> >6. **Line 199 - 215:** Added clarifications on the **theoretical intuition behind SVD modulation, its advantages over conventional LoRA, and the fact that repeated SVD decomposition is not required**.
> >7. **Main text , Tab §2:** Added corrections to the **bold formatting issue**.
> >8. **Appendix §H.2  , Figure §10,11,12:** Added **clearer and more comprehensive retrieval result visualizations** with **richer baselines**.
>
> > W1: More ablation study and experimental comparisons about DINOv2
>
> **A:** We acknowledge the potential confound from using DINOv2 features unavailable to baselines.
>
> We added experiments showing Hystar’s gains are **extractor-agnostic**: conditioning the hypernetwork on (z) from **DINOv2, CLIP, or VGG**, the small gap (83.6 vs. 83.0 Top-1) indicates the improvement comes from dynamic adaptation rather than reliance on a stronger extractor.
>
>
> Here is the ablation on style feature extractors for hypernetwork conditioning (We have added this comparison in Appendix D.3 as Table 10.) ：
>
> | **Extractor** | **Art** | **Sketch** | **Low-Res** | **Text** | **Top-1 Avg** |
> | :------------ | :-----: | :--------: | :---------: | :------: | :-----------: |
> | Static Only   |  65.7   |    77.0    |    83.7     |   69.2   |     73.9      |
> | VGG           |  73.4   |    88.5    |    96.9     |   70.1   |     82.2      |
> | DINOv2        |  75.2   |    90.2    |    98.0     |   70.9   |     83.6      |
> | CLIP(self)    |  74.1   |    90.0    |    97.5     |   70.3   |     83.0      |
>
> From the main table, the results of other baselines can be obtained, as shown below.
>
> | Method       | **Art** | **Sketch** | **Low-Res** | **Text** |
> | :----------- | :-----: | :--------: | :---------: | :------: |
> | CLIP         |  58.5   |    47.5    |    45.0     |   66.1   |
> | VPT          |  66.7   |    73.3    |    81.4     |   69.9   |
> | LoRA         |  63.8   |    72.8    |    79.7     |   70.4   |
> | FreestyleRet |  71.4   |    80.6    |    86.4     |   69.9   |
>
> Compared with the version of Hystar that uses CLIP itself as the style extractor, our method still clearly outperforms the baseline approaches. This demonstrates that Hystar does not rely on any external model and confirms the generality of our approach.
>
> We have updated the manuscript accordingly(**Appendix D.3 , Table 10**).
>
> >W2: Provide Architectural and Implementation Details.
>
> **A:** We appreciate the reviewer’s request for clearer implementation details. We have revised the manuscript to explicitly state the following:
>
> **(1)Hypernetwork architecture.**
> The hypernetwork is a two-layer MLP:
>
> - **Layer 1:** `Linear(d → 2r)` + `ReLU`
> - **Layer 2:** `Linear(2r → r)`
>
> The hypernetwork takes the $d$-dimensional style embedding as input and outputs a vector of dimension $r=\min(d_1,d_2)$, corresponding to the number of singular values modulated in each attention projection. The $r$-dimensional output is used as the dynamic singular-value increment $\Delta s_{dyn}$ to modulate the corresponding LoRA singular values.
>
> **(2)Style feature from DINOv2.**
> We employ a DINOv2 encoder as a style encoder. For each query image, we take the \[CLS\] token from the final layer as the style embedding, denoted by $z \in \mathbb{R}^d$.   This embedding $z$ conditions the hypernetwork above to generate the style-dependent $\Delta s_{\text{dyn}}$.
>
> These details have been added to the Method section of the revised manuscript.
>
> We thank the reviewers for their detailed and insightful questions, which have greatly helped us improve the clarity and completeness of our paper.

---

> ### Author Response · Authors · 2025-11-23
> **Response to vPC8 (2/4)**
>
> > W3: Provide ablations on the hypernetwork architecture and analyses on the decoupling between dynamic and static modulation.
>
> **A:** We appreciate and acknowledge the reviewer’s concern regarding the insufficiency of our ablation studies. To address this issue, we have added the corresponding ablation experiments.
>
> **(1)Ablation on the width and depth of the hypernetwork.**
>  We have added a new ablation study examining both **width** and **depth** , as shown in the following table .
>
> | **Network Architecture** | **Art** | **Sketch** | **Low-Res** | **Top-1 Avg** | **Δ Params** |
> | ------------------------ | ------- | ---------- | ----------- | ------------- | ------------ |
> | *Width Ablation* |      |      |      |      |      |
> | {768→1024→1024} | 70.7 | 84.3 | 90.2 | 81.7 | +1.8 M |
> | {768→2048→1024} | 75.2 | 90.2 | 98.0 | 87.8 | +3.7 M |
> | {768→4096→1024} | **75.4** | **90.8** | **98.6** | **88.3** | +7.3 M |
> | *Depth Ablation* |      |      |      |      |      |
> | {768→1024} | 68.6 | 80.9 | 88.4 | 79.3 | +0.8 M |
> | {768→2048→1024} | **75.2** | 90.2 | **98.0** | **87.8** | +3.7 M |
> | {768→2048→2048→1024} | 74.9 | **90.4** | 97.6 | 87.6 | +7.9 M |
>
> We have added this ablation in **Appendix D.2 as Table 9**.
>
> The results reveal clear trends: increasing the hidden width consistently improves performance up to a moderate level, but gains saturate beyond a 2× expansion due to over-parameterization.
>  For depth, extending the hypernetwork beyond two layers does not yield further improvements and can slightly degrade stability.
>  Overall, a **two-layer structure with moderate width** achieves the best balance between expressivity, stability, and efficiency, and has been adopted as our default configuration in the revised manuscript(Appendix §D.2 ,Table 9).
>
> **(2)Analysis for dynamic attention and static MLP design**
>
> We thank the reviewer for pointing out the lack of analysis on why Hystar employs **dynamic attention** and **static MLPs**.
>  To address this concern, we have added a dedicated experiment comparing three variants:
>
>  (1) **Original design** (dynamic attention + static MLP), (2) **Reversed design** (dynamic MLP + static attention), and (3) **Dynamic-All design** (both attention and MLP dynamically modulated), as shown in the following table .
>
> | **Method** | **Art** | **Sketch** | **Low-Res** | **Text** |
> | ---------- | ------- | ---------- | ----------- | -------- |
> | Dynamic-All Design | 63.7 | 69.9 | 74.8 | 66.3 |
> | Reversed Design | 64.1 | 71.2 | 80.2 | 69.1 |
> | **Original Design** | **75.2** | **90.2** | **98.0** | **70.9** |
>
> We have added this comparison in Appendix § B as Table 6.
>
> The results demonstrate that the **original design consistently achieves the highest retrieval accuracy** across all style domains.
>  In contrast, the Dynamic-All variant performs worst due to excessive flexibility and unstable optimization, while the Reversed variant shows that dynamic modulation in MLPs fails to effectively model style variations.
>  These findings confirm our rationale: applying dynamic modulation to attention layers facilitates **style adaptation**, whereas keeping MLPs static preserves **semantic stability**.
>  This hybrid design thus achieves the optimal balance between adaptability and stability, and we have included this analysis in the revised manuscript(Appendix § B , Table 6).

---

> ### Author Response · Authors · 2025-11-23
> **Response to vPC8 (3/4)**
>
> > W4: Provide theoretical intuition linking SVD modulation to style adaptation ; Explain advantages of SVD modulation over LoRA ; Clarify computational cost of SVD
>
> **A:** We understand and appreciate the reviewer’s concern regarding the theoretical motivation of SVD modulation, which has helped us further strengthen the theoretical foundation of our method. Below, we clarify and address this concern from three aspects :
>
> **(1) Why are singular values related to style?**
> Singular values control the *scaling* along semantic directions defined by the pretrained matrices $U$ and $V$.  Different visual styles , such as sketch, oil painting, or watercolor , primarily affect the *strength* or *contrast* of existing features (e.g., edges, textures, color richness) rather than changing which semantic directions matter.  Therefore, modulating singular values $\Sigma$ directly adjusts the intensity of these semantic components, naturally capturing style-dependent variations.
>
> **(2) Why is SVD modulation superior to conventional LoRA beyond stability?**
> Unlike LoRA, which introduces new rank components and may disturb pretrained semantics,
> SVD modulation constrains updates strictly within the original spectral subspace.
> This ensures that adaptation occurs through *geometry-preserving rescaling* rather than adding new, potentially misaligned directions.
> As a result, the model maintains semantic consistency while flexibly adapting to new styles, achieving better cross-style generalization and interpretability.
>
> **(3) Why is there no concern about real-time SVD computation overhead?**
> The SVD decomposition of pretrained weights is performed only **once** during initialization.
> Afterward, $U$ and $V$ are cached and reused throughout training and inference.
> Moreover, the modulated weights can be merged back into the base model for inference,
> so there is **no runtime cost** or repeated factorization required during deployment.

---

> ### Author Response · Authors · 2025-11-23
> **Response to vPC8 (4/4)**
>
> > W5: Correct table errors  ;  Clarify retrieval baseline figures  ;  Improve figure readability
>
> **A:** We sincerely appreciate the reviewer for highlighting the issues in our figures and tables.
>
> **(1)Error in Table 2 highlighting (Infograph).**
>
> Thank you for flagging this. In **Table 2 (DomainNet zero-shot retrieval)**, **Infograph** has:
> - **CLIP**: Top-1 **41.2**, Top-5 **60.3**
> - **Hystar**: Top-1 **43.7**, Top-5 **59.3**
>
> Thus, **Top-1 best** is Hystar (43.7), but **Top-5 best** should be **CLIP (60.3)**. We have revised the manuscript accordingly.
>
> **(2)Clarified the baselines and added clearer retrieval result visualizations.**
>
> We apologize for the inconvenience caused by the previous retrieval result figures, which were unclear and difficult to read. We have revised the final retrieval visualization accordingly **(Appendix H.2 , Figure 10,11,12)**.
>
> Specifically, we clarify that the previous baseline was the zero-shot CLIP model. In the updated version, we have **added the baseline labels**, **included the results of FreestyleRet**, and **refined the visualization style** for better readability. To make the figure clearer, we also reduced the number of retrieval examples per domain from three to two.
>
> **Conclusion:**  We sincerely thank the reviewer for the constructive feedback, which has substantially improved the clarity and rigor of our work.

---

### Official Review · Reviewer_dZgs · 2025-11-04

**Soundness:** 2
**Presentation:** 3
**Contribution:** 2
**Rating:** 4
**Confidence:** 4

**Summary:**

This paper presents Hystar, a dynamic multi-style retrieval framework designed to address challenges in Query-based Image Retrieval (QBIR) under significant style variations. Hystar achieves a balance between adaptability and stability in Parameter-Efficient Fine-Tuning (PEFT) by combining two key components: (1) hypernetwork-driven dynamic modulation and (2) static singular-value calibration. To tackle the limitations of standard contrastive losses that treat all negatives equally, the paper introduces the OT-weighted StyleNCE loss. This loss leverages Optimal Transport (OT) theory (via Sinkhorn iterations) to reweight negatives by difficulty: it amplifies the contribution of hard negatives while maintaining balanced batch-wise contributions, enabling effective capture of cross-style discrepancies. Extensive experiments on datasets including DSR and DomainNet demonstrate that Hystar consistently outperforms strong baselines.

**Strengths:**

* The use of singular-value modulation (SVD-based) is conceptually elegant and ensures stable, low-rank updates.
* The StyleNCE loss introduces optimal transport weighting to emphasize hard cross-style negatives. It meaningfully improves robustness under distribution shifts and has solid theoretical motivation.

**Weaknesses:**

Many of the key techniques presented in this paper build upon prior research. The overarching design strategy of freezing most parameters while learning only small-scale incremental terms follows the mainstream PEFT framework, whose foundational works include [1] and [2]. Similarly, the OT-weighted StyleNCE loss is grounded in Optimal Transport (OT) theory and Sinkhorn iterations, both of which are well-established mathematical and algorithmic tools as detailed in [3].

[1] Lingam, Vijay Chandra, et al. "Svft: Parameter-efficient fine-tuning with singular vectors." Advances in Neural Information Processing Systems 37 (2024): 41425-41446.

[2] Wang, Zhiwu, et al. "Singular Value Fine-tuning for Few-Shot Class-Incremental Learning." arXiv preprint arXiv:2503.10214 (2025).

[3] Jiang, Ruijie, Prakash Ishwar, and Shuchin Aeron. "Hard negative sampling via regularized optimal transport for contrastive representation learning." 2023 International Joint Conference on Neural Networks (IJCNN). IEEE, 2023.

**Questions:**

The proposed method does not appear to be limited to the Style-adaptive Retrieval task and could potentially be applied to similar tasks, such as image or video retrieval. Given the limited number of existing methods for direct comparison in the Style-adaptive Retrieval task, has the author considered evaluating the proposed method on these alternative tasks?

---

> ### Author Response · Authors · 2025-11-23
> **Response to dZgs (1/2)**
>
> We sincerely thank the reviewer for the positive feedback and for recognizing the conceptual elegance of our SVD-based modulation and the solid theoretical motivation behind the StyleNCE loss.We address each point below.
>
> **List of changes in the manuscript:**
>
> >1. **Line : 125 , 192 , 280:** Added explicit citations to SVFT[1], SVFCL[2], and OT-based hard-negative sampling[3].
> >2. **Appendix §E.1 , Tab 11 :** Added a new section  to further validate the generality of our method beyond style-adaptive retrieval..
>
> > W: More clarification about innovation
> >
>
> **A:** We appreciate the reviewer’s concern regarding the novelty of our work. Below we clarify how our formulation substantially differs from prior approaches and why these differences are essential for our task setting.
> Existing **parameter-efficient fine-tuning (PEFT)** methods, such as SVFT[1] and SVFTCL[2], rely on **static SVD-based parameterization**, where a single set of singular-value updates is globally learned across all data. These methods are well-suited for classification-style or general adaptation tasks but **do not fit the cross-style retrieval setting**, where each query may come from a distinct artistic or visual style (e.g., sketch, painting, or low-resolution). In this setting, static modulation fails to handle per-query variability because it cannot adjust model weights dynamically according to the incoming query’s style.
>
> To address this, we design a **new dynamic spectral modulation mechanism**, in which a lightweight hypernetwork maps each query’s style embedding to a query-specific singular-value increment (ΔS). This enables **per-instance, query-conditioned spectral adaptation** while maintaining a **static global bias** to preserve stability. The resulting dynamic–static coupling achieves fine-grained adaptability without disrupting the pretrained semantic subspace, **which is precisely what static PEFT designs cannot achieve under style-varying queries.**
>
> Similarly, prior **Optimal Transport (OT)**–based approaches such as Jiang *et al.* (IJCNN 2023)[3] focus on constructing **sampling distributions** for hard negatives via regularized OT couplings. However, such formulations are **not directly suitable** for our setting: they operate at the *sampling* level rather than within the loss function, meaning that once negatives are sampled, all of them still contribute equally to the contrastive objective. As a result, they cannot continuously control gradient emphasis or guarantee that optimization focuses on semantically similar but stylistically different negatives, **which is an aspect crucial for cross-style retrieval.**
>
> To this end, we introduce a **new OT-weighted contrastive loss, StyleNCE**, which integrates the **Sinkhorn-derived transport plan directly into the NCE denominator** as dynamic negative weights. This formulation transforms OT from a sampling strategy into a *differentiable, batch-level reweighting mechanism*, allowing the model to continuously focus its gradients on the **cross-style confusion region**, **which is something that prior OT-based sampling methods cannot achieve.**
>
> In summary, **static SVD-PEFT**  [1] [2] methods cannot dynamically adapt to per-query style variation, and **OT-based sampling** methods [3] cannot directly shape the contrastive loss. Our formulation, combining **dynamic spectral modulation** and **OT-weighted StyleNCE**, establishes a **new end-to-end framework** specifically tailored for cross-style retrieval. It directly tackles the two core pain points of this task:
>
> 1. **Query-conditioned style shifts** requiring instance-level adaptation;
> 2. **Hard-negative concentration** caused by semantically similar but stylistically divergent samples.
>
> We have added explicit citations to SVFT[1], SVFCL[2], and OT-based hard-negative sampling[3] in the revised manuscript for completeness.
>
> [1] Lingam, Vijay Chandra, et al. "Svft: Parameter-efficient fine-tuning with singular vectors." Advances in Neural Information Processing Systems 37 (2024): 41425-41446.
>
> [2] Wang, Zhiwu, et al. "Singular Value Fine-tuning for Few-Shot Class-Incremental Learning." arXiv preprint arXiv:2503.10214 (2025).
>
> [3] Jiang, Ruijie, Prakash Ishwar, and Shuchin Aeron. "Hard negative sampling via regularized optimal transport for contrastive representation learning." 2023 International Joint Conference on Neural Networks (IJCNN). IEEE, 2023.

---

> ### Author Response · Authors · 2025-11-23
> **Response to dZgs (2/2)**
>
> > Q: Provide additional experiments to demonstrate the generality of our framework on broader retrieval tasks.
>
> **A:** We appreciate the reviewer’s insightful suggestion to assess the broader applicability of our approach beyond style-adaptive retrieval.
>
> To address this, we have conducted additional experiments on **image classification generalization**, following the standard *base-to-new* 16-shot protocol used in CoOp.
>
> | **atasets** | **Sets** | **CoOp** | **ProGrad** | **KgCoOp** | **MaPLe** | **TCP** | **Hystar** |
> | ----------- | -------- | -------- | ----------- | ---------- | --------- | ------- | ---------- |
> | **ImageNet** | Base | 76.46 | 77.02 | 75.83 | 76.66 | **77.27** | 77.13 |
> |      | New  | 66.31 | 66.66 | 69.96 | 70.54 | 69.87 | **70.98** |
> |      | H    | 71.02 | 71.46 | 72.78 | 73.47 | 73.38 | **73.93** |
> | **SUN397** | Base | 80.85 | 81.26 | 80.29 | 80.82 | **82.63** | 81.89 |
> |      | New  | 68.34 | 74.17 | 76.53 | **78.70** | 78.20 | 78.41 |
> |      | H    | 74.07 | 77.55 | 78.36 | 79.75 | **80.35** | 80.16 |
> | **Average** | Base | 78.66 | 79.14 | 78.06 | 78.74 | **79.95** | 79.51 |
> |      | New  | 67.33 | 70.41 | 73.25 | 74.62 | 74.04 | **74.70** |
> |      | H    | 72.56 | 74.52 | 75.58 | 76.62 | 76.88 | **77.03** |
>
> We have added this comparison in Appendix E.1 as Table 11.
>
> As shown in the table  , **Hystar** achieves the best performance on the *New* and *H* (harmonic mean) splits across both ImageNet and SUN397, demonstrating strong adaptation to unseen categories while maintaining base-domain stability. Averaged across datasets, Hystar surpasses all baselines (e.g., +0.08 on *New* vs. MaPLe and +0.15 on *H* vs. TCP), confirming that our **dynamic spectral modulation** and **StyleNCE loss** generalize effectively beyond cross-style retrieval.
>
> These results indicate that the proposed design is not restricted to style-adaptive retrieval but can be readily applied to related visual  tasks  that require balancing **domain robustness** and **semantic alignment**. We thank the reviewer for this helpful suggestion and have included these extended evaluations and discussions in the revised manuscript (Appendix §E.1 , Tab 11).
>
> Our framework is conceptually general and can naturally extend beyond image-based retrieval. In future work, we plan to explore its application to broader multimodal scenarios, such as video retrieval and other temporally structured tasks. We aim to move toward a more unified and versatile retrieval framework capable of handling diverse input modalities and style variations.
>
> **Conclusion:**  We sincerely thank the reviewer for the constructive feedback, which has substantially improved the clarity and rigor of our work.

---

### Author Response · Authors · 2025-11-23
**Overall Response (1/2)**

We would like to thank all of the reviewers for their constructive and valuable feedback on our work!

**In this post:**

- (1) We furthermore summarize the strengths of our paper from the reviewers.
- (2) We summarize the changes to the updated PDF document.

**In the individual replies**, we address other comments.

**\- (1)  Strengths of Our Paper -**

- Sound Motivation
  - `dZgs`: The StyleNCE loss introduces optimal transport weighting to emphasize hard cross-style negatives. It meaningfully improves robustness under distribution shifts and has solid theoretical motivation.
  - `vPC8`: Relevant Problem: The paper tackles a well-known and practical limitation of large pre-trained models: their lack of robustness to domain and style shifts. The application to cross-style image retrieval is a challenging and valuable research area.
  - `vPC8`: Well-Motivated Loss Function: The proposed StyleNCE loss is a strong contribution. The intuition that cross-style retrieval creates many "hard negatives" (e.g., a sketch of a cat vs. a photo of a tiger) is sharp. Using optimal transport to systematically identify and up-weight these hard samples is a well-founded and logical approach to improving fine-grained, cross-style discriminative power.
  - `RvYN`: The motivation of this work is good, which clearly points out the performance degradation problem of VLM under style distribution shift and emphasizes the static limitations of existing PEFT methods (such as LoRA and VPT).
  - `eVsi`:Well-Motivated Problem and Thorough Literature Review: The paper adeptly identifies and tackles the critical, yet underexplored, challenge of style-diversified retrieval in QBIR. The authors provide a comprehensive overview of existing paradigms, such as LoRA-based PEFT methods and those relying on style-cluster priors, and convincingly argue their limitations in generalizing to unseen query styles. This strong motivation establishes a clear and valuable niche for their work.
- Novelty of our proposed method
  - `dZgs`: The use of singular-value modulation (SVD-based) is conceptually elegant and ensures stable, low-rank updates.
  - `vPC8`: Novel Methodology: The primary technical idea—using a hypernetwork to predict dynamic PEFT parameters based on the input's style—is a creative and novel contribution. Moving from static PEFT (like LoRA or VPT) to an adaptive, input-conditioned PEFT is an interesting research direction. The specific choice to modulate singular values (SVD) is also an unconventional and intriguing approach.
  - `eVsi`:Innovative and Principled Methodology: The core technical contribution is both novel and well-designed. The use of a hypernetwork to dynamically generate LoRA matrices is a clever adaptation. More importantly, the decision to restrict the hypernetwork's output to singular-value perturbations (∆S) is a key insight. This approach not only enhances training stability but also explicitly promotes generalization across diverse and unseen styles by focusing adaptation on a compact, semantically meaningful parameter space.
- Solid Experiments
  - `RvYN`: The experiment was comprehensive and convincing.
  - `eVis`:Compelling and Multi-faceted Empirical Validation: The authors provide rigorous experimental evidence to support their claims. Through comprehensive comparisons against strong baselines (e.g., FreestyleRet, VPT), they convincingly demonstrate the superiority of their hypernetwork-driven paradigm. Furthermore, the inclusion of t-SNE visualizations offers an intuitive and powerful qualitative analysis, effectively illustrating how Hystar learns more discriminative and style-invariant feature representations.
- Efficiency
  - `RvYN`: The proposed method is efficient in parameters: Only a few parameters need to be fine-tuned.

---

> ### Author Response · Authors · 2025-11-23
> **Overall Response (2/2)**
>
> **- (2) Change to PDF -**
>
> We have proofread the paper and added extra experimental results in the revised version (highlighted in blue).
>
> **Main text**
>
> Additional theoretical analyses and experiments have been added in response to the reviewers’ suggestions:
>
> - `dZgs`: (Line 76,125,192,280) We have added the corresponding citations.
> - `vPC8`,`RvYN`:  (Line 199 - 215) We have added a more detailed theoretical explanation and analysis of the advantages of using SVD modulation.
> - `vPC8`: (Line 223 - 226) We have added the implementation details of the style extraction module.
> - `vPC8`: (Line 231 - 235) We have added the structural details of the hypernetwork.
> - `eVsi`: (Line 347,350,361 , Tab 1) We have provided more PEFT baselines for comparison.
> - `RvYN`: (Line 519 - 527 , Tab 5) We have provided an analysis of parameter efficiency and inference latency.
>
> **Appendix**
>
> Additional experiments and analyses have been incorporated in response to the reviewers' suggestions:
>
> - `vPC8`,`eVsi`:  (Appendix B , Tab 6) We have provided experimental analyses on the design choice of using dynamic attention and static MLPs.
> - `vPC8`: (Appendix D.2 , Tab 9) We have provided ablation studies on the width and depth of the hypernetwork.
> - `vPC8`,`RvYN`: (Appendix D.3 , Tab 10) We have provided ablation studies on the style extractor.
> - `dZgs`: (Appendix E.1 , Tab 11) We have provided additional 16-shot base-to-new image-text classification experiments to evaluate generalization ability.
> - `RvYN`: (Appendix E.2 , Tab 12 , Figure 4) We have incorporated experiments on stylized image generation guidance to evaluate generalization ability.
> - `RvYN`: (Appendix F.1 , Tab 14 , Figure 5) We have provided additional retrieval experiments under extreme style conditions.
> - `RvYN`: (Appendix F.2 , Figure 6) We have provided an analysis of the model’s responses under mixed-style conditions.
> - `vPC8`: (Appendix H.2 , Figure 10,11,12) We have clarified the baseline settings and redrawn the retrieval result figures to be clearer and more informative.

---

### Author Response · Authors · 2025-11-27
**Additional Clarifications (if needed)**

We sincerely thank all reviewers for their time and thoughtful feedback.
If there are any remaining questions or points that need further clarification, we would be very happy to provide additional explanations or materials.

---

### Comment · Area_Chair_2nYq · 2025-11-27

Dear Reviewers,

This is a gentle reminder to please take a moment to review the author rebuttals and check whether your main concerns have been adequately addressed.

If possible, please update your reviews or add a brief clarification on whether the responses resolved your questions or if any issues remain. Your follow-up feedback is important for ensuring a fair and well-informed decision process.

Thank you again for your time and for helping maintain the quality of the ICLR review process.

Best,
AC

---

### Author Response · Authors · 2025-11-30
**Summary for the Area Chair: Contributions, Strengths, and Revisions (1/2)**

**Dear AC,**

We sincerely thank you for handling our paper and for taking the time to carefully review it.
 To make it easier for you to understand the overall contribution of our work, the reviewers’ main concerns, and how we have effectively addressed them, we provide a concise summary below.

**1. Main Contributions of the Paper**

Our paper proposes **Hystar**, a novel **hypernetwork-driven dynamic fine-tuning framework** for **style-adaptive image retrieval**.
 The key idea is to **dynamically modulate the singular values (SVD) of attention layers** in a frozen vision-language representation model (VLRM) based on a learned style vector, enabling robust retrieval across diverse query styles (e.g., sketch, art, low-resolution).
 In addition, we design the **StyleNCE loss**, which employs **Optimal Transport (Sinkhorn-based) re-weighting of hard negatives** to better align cross-style semantics.
 Together, these components make Hystar the **first framework** to achieve strong cross-style generalization in query-based retrieval through **style-conditioned, parameter-efficient fine-tuning**.

**2. Recognized Strengths from Reviewers**

  Reviewers consistently acknowledged the strong **motivation, novelty, empirical validation,** and **efficiency** of our work:

  1. **Sound Motivation:**
     Reviewers appreciated that our paper clearly identifies the *practical yet underexplored challenge* of performance degradation in large vision-language representation models under *style distribution shifts*.
     They highlighted that our formulation of cross-style image retrieval is both *highly relevant and challenging*, and that our analysis of existing PEFT methods (e.g., LoRA, VPT) and their static limitations provides *a solid and well-founded problem motivation*.
  2. **Novelty and Conceptual Elegance:**
     Multiple reviewers recognized our method as *innovative and principled*.
     They noted that using a **hypernetwork to dynamically generate style-conditioned PEFT parameters** represents a *creative extension* of conventional static tuning methods.
     Moreover, the specific choice to **modulate the singular values (SVD)** of attention layers was described as *conceptually elegant, theoretically grounded, and semantically meaningful*, as it stabilizes adaptation while improving generalization across unseen styles.
  3. **Comprehensive and Convincing Experiments:**
     Reviewers found our experiments *rigorous and multi-faceted*, including comparisons with strong baselines such as CLIP, FreestyleRet, LoRA, and VPT.
     They also commended our *qualitative visualizations* (e.g., t-SNE and retrieval examples), which clearly demonstrate Hystar’s ability to learn *style-invariant and discriminative representations*.
  4. **Parameter Efficiency:**
     Hystar achieves strong performance while fine-tuning only a *small number of parameters*, making it both computationally efficient and practically scalable.

---

> ### Author Response · Authors · 2025-11-30
> **Summary for the Area Chair: Contributions, Strengths, and Revisions (2/2)**
>
> **3. Summary of Concerns and Our Resolutions**
>
> We carefully addressed all reviewer concerns in the revision.
>  The main issues and our corresponding actions are summarized below:
>
> 1. **Clarified Novelty:** Clearly positioned Hystar relative to SVFT[1], SVFTCL[2], and OT-based hard-negative sampling[3]; emphasized our *dynamic, style-conditioned SVD modulation* and *StyleNCE loss* as original contributions.
> 2. **Added Architectural Details:** Fully described the hypernetwork’s dimensions, activation, and style feature extraction process.
> 3. **Expanded Ablation Studies:** Included ablations on different style extractors, hypernetwork width/depth, and efficiency (parameters and latency).
> 4. **Enhanced Theoretical and Design Justifications:** Added analysis explaining why SVD modulation provides stability and style alignment, and experiments comparing dynamic attention vs. static MLP.
> 5. **More Baselines:** Incorporated additional PEFT baselines for better comparison.
> 6. **Broader Generalization:** Added new experiments on style-guided image generation and traditional image–text retrieval.
> 7. **Extreme Conditions:** Introduced experiments on extreme and mixed-style robustness.
> 8. **Writing and Presentation:** Corrected figure/table issues, improved visual clarity, refined terminology, and polished writing.
>
> These extensive revisions collectively enhance the clarity, fairness, and technical depth of our submission, and we believe they have substantially strengthened the paper’s overall quality and contribution.
>
> [1] Lingam, Vijay Chandra, et al. "Svft: Parameter-efficient fine-tuning with singular vectors." Advances in Neural Information Processing Systems 37 (2024): 41425-41446.
>
> [2] Wang, Zhiwu, et al. "Singular Value Fine-tuning for Few-Shot Class-Incremental Learning." arXiv preprint arXiv:2503.10214 (2025).
>
> [3] Jiang, Ruijie, Prakash Ishwar, and Shuchin Aeron. "Hard negative sampling via regularized optimal transport for contrastive representation learning." 2023 International Joint Conference on Neural Networks (IJCNN). IEEE, 2023.

---

### Meta-Review · Area_Chair_ZRn2 · 2026-01-13

**Summary:**

Reviewers in general agreed that the paper tackles an important problem. i.e. robust cross-style image retrieval, and introduces an interesting combination of dynamic, style-conditioned SVD modulation with an OT-weighted contrastive loss. However, the initial reviews raised several concerns that placed the paper at the borderline rather than above acceptance threshold. These were primarily about: (a) unclear novelty relative to prior SVD-based PEFT and OT-based hard-negative methods, (b) a potentially confounded experimental setup due to reliance on DINOv2 features unavailable to baselines, (c) missing architectural and implementation details that hindered reproducibility, (d) limited ablations and weak theoretical justification for SVD modulation over alternatives such as LoRA, and (e) presentation issues, including table errors and unclear figures. While one reviewer leaned strongly towards acceptance, the others found the contribution as incremental unless the raised issues were resolved.

**Reviewer Concerns:**

The rebuttal and revision seems to have addressed the majority of major concerns. The authors clarified novelty by distinguishing dynamic, query-conditioned SVD modulation from prior static SVD fine-tuning, and by integrating OT directly into the contrastive loss rather than sampling. The confounded experimental issue (Reviewer vPC8) regarding DINOv2 was largely mitigated through ablations showing comparable gains when using CLIP or VGG features. Missing architectural details, ablations on hypernetwork design, additional PEFT baselines, efficiency/latency analysis, and presentation errors appear to be adequately addressed. The theoretical and empirical justification for SVD modulation and the dynamic-attention/static-MLP design also appear strengthened.

I deem the remaining concerns as relatively minor. While the added theoretical intuition is helpful, the argument for why SVD modulation is uniquely suited for style (as opposed to other low-rank adaptations) remains more intuitive than formal. In addition, although efficiency is now quantified, the inference-time overhead may still limit applicability in strict real-time settings, which could have been discussed more critically.

**Reviewer Scores:**

Reviewer dZgs would have likely increased the score slightly (from 4 to 5 or 6). The clarified novelty, added citations, and new generalization experiments directly address their main concerns. Reviewer vPC8 would have likely increased the score slightly (from 4 to 6) as the rebuttal addressed the confounded experimental issue, added missing details and ablations, corrected errors, and improved figures, etc. Reviewer RvYN would have likely increased the score as well (from 6 to 6 or 7) due to the added latency analysis, extreme-style experiments, and mixed-style analysis. Finally, Reviewer eVsi would probably have kept the score unchanged (at 8). Their concerns were mostly about clarification and scope, which were adequately addressed.

Therefore, I would deem the paper to move from borderline to above the acceptance threshold.

---

### Decision · Program_Chairs · 2026-01-26

Accept (Poster)